# Mechanism of allosteric regulation of β₂-adrenergic receptor by cholesterol

Moutusi Manna[1], Miia Niemelä[1], Joona Tynkkynen[1], Matti Javanainen[1,2], Waldemar Kulig[1,2], Daniel J Müller[3], Tomasz Rog[1,2], Ilpo Vattulainen[1,2,4*]

[1]Department of Physics, Tampere University of Technology, Tampere, Finland; [2]Department of Physics, University of Helsinki, Helsinki, Finland; [3]Department of Biosystems Science and Engineering, ETH Zürich, Basel, Switzerland; [4]MEMPHYS-Center for Biomembrane Physics, University of Southern Denmark, Odense, Denmark

**Abstract** There is evidence that lipids can be allosteric regulators of membrane protein structure and activation. However, there are no data showing how exactly the regulation emerges from specific lipid-protein interactions. Here we show in atomistic detail how the human β₂-adrenergic receptor (β₂AR) – a prototypical G protein-coupled receptor – is modulated by cholesterol in an allosteric fashion. Extensive atomistic simulations show that cholesterol regulates β₂AR by limiting its conformational variability. The mechanism of action is based on the binding of cholesterol at specific high-affinity sites located near the transmembrane helices 5–7 of the receptor. The alternative mechanism, where the β₂AR conformation would be modulated by membrane-mediated interactions, plays only a minor role. Cholesterol analogues also bind to cholesterol binding sites and impede the structural flexibility of β₂AR, however cholesterol generates the strongest effect. The results highlight the capacity of lipids to regulate the conformation of membrane receptors through specific interactions.

*For correspondence: Ilpo. Vattulainen@helsinki.fi

Competing interests: The authors declare that no competing interests exist.

## Introduction

G protein-coupled receptors (GPCRs) are versatile signaling proteins that mediate diverse cellular responses. With over 800 members, GPCRs constitute the largest family of integral membrane proteins in human genome and represent roughly half of all drug targets in modern medicine (*Gilchrist, 2010*).

The human β₂-adrenergic receptor (β₂AR) is one of the best-characterized GPCRs. It is expressed in pulmonary and cardiac myocyte tissues and is a therapeutic target for asthma and heart failure (*Lefkowitz, 2000*). The functional diversity of β₂AR is associated with its structural dynamics (*Manglik and Kobilka, 2014*; *Kobilka, 2013*). Recently found structures of β₂AR in the inactive and active states have provided valuable insights into the structure-function relationship of β₂AR (*Cherezov et al., 2007*; *Hanson et al., 2008*; *Rasmussen et al., 2011*). Subsequent biophysical and biochemical studies have provided direct evidences of multiple distinct conformational states for specific GPCRs, such as β₂AR (*Manglik and Kobilka, 2014*; *Kobilka, 2013*; *Nygaard et al., 2013*). Meanwhile, molecular dynamics (MD) simulations have depicted the dynamic behavior of β₂AR and have significantly enhanced our understanding of the activation mechanism of GPCRs (*Dror et al., 2009*; *Ozcan et al., 2013*; *Dror et al., 2011*). Intriguingly, it is now evident that the activation of GPCRs is modulated by lipids (*Oates and Watts, 2011*).

The lipid raft concept (*Lingwood and Simons, 2010*; *Allen et al., 2007*) essentially states that cell membranes include functional nanoscale domains where the function emerges from proteins whose structure and activation are modulated by lipids. However, despite a large body of research

**eLife digest** Proteins known as G protein-coupled receptors, or GPCRs for short, detect and respond to hormones and other signaling molecules found outside cells. A signaling molecule activates a GPCR by binding to it and causing the receptor to change its shape. This triggers a cascade of signals inside the cell that leads to the cell responding in a particular way. There are over 800 different GPCRs in human cells, making them the largest family of cell surface proteins.

GPCRs span the membrane that surrounds each cell. This membrane is made of molecules called lipids and previous studies have shown that many lipids are able to bind to GPCRs and influence their shape and activity. Lipids can cause these changes via so-called 'allosteric' regulation, in which the lipid binds to a site on the receptor that is separate to where the signal molecule binds. Lipid binding can either enhance or inhibit the activity of the receptor.

Human $\beta_2$-adrenergic receptor is one of the best-studied GPCRs. It responds to a hormone called epinephrine (also known as adrenaline), which plays important roles in many organs in the body, including the heart and lungs. A lipid called cholesterol, which is plentiful in the cell membrane, can also bind to this receptor and influence its shape, but how this happens was not fully understood. Manna et al. now use computer simulations to analyze the interaction between cholesterol and $\beta_2$-adrenergic receptor in more detail.

The simulations reveal that cholesterol makes the $\beta_2$-adrenergic receptor less flexible so that it can only adopt certain shapes. This helps to stabilize both the inactive and active states of the receptor so that it is not as easy for the receptor to switch between them. The cholesterol molecules bind to specific sites on the receptor within the region of the protein that crosses the cell membrane.

The new findings of Manna et al. provide detailed insights into how cholesterol governs the shape and activity of the $\beta_2$-adrenergic receptor. The next step is to extend this analysis to other types of lipids and GPCRs.

data, direct substantiation of lipid-induced protein modulation remains limited. Contreras et al. showed that the COPI machinery protein p24 is recognized by a specific sphingomyelin (*Contreras et al., 2012*). Coskun et al. showed that monosialoganglioside GM3 influences the activation of the epidermal growth factor receptor (*Coskun et al., 2011*), however the mechanism is not known. Lipid modulation also holds to GPCRs (*Oates and Watts, 2011*; *Neale et al., 2015*; *Dawaliby et al., 2016*) in particular through cholesterol (*Oates and Watts, 2011*; *Paila and Chattopadhyay, 2009*; *Gimpl et al., 1997*; *Paila et al., 2011*; *Muth et al., 2011*), which changes the physical properties of cellular membranes and supports the dynamic assembly of nanoscale membrane domains (*Simons and Ikonen, 2000*).

The best known case is $\beta_2$AR, which is a prototype of cholesterol-interacting GPCRs. $\beta_2$AR belongs to the family of class A GPCRs. GPCRs belonging to this class show a high structural similarity and functional diversity. The literature reporting on the specific functional role of cholesterol and other lipids is extensive (*Pucadyil and Chattopadhyay, 2006*; *Gimpl, 2016*). It has been experimentally shown that cholesterol affects the conformation (*Muth et al., 2011*; *Casiraghi et al., 2016*) and function (*Gimpl et al., 1997*; *Paila et al., 2011*; *Pucadyil and Chattopadhyay, 2006*; *Casiraghi et al., 2016*; *Jafurulla et al., 2014*) of many GPCRs. Based on X-ray crystal structures cholesterol has specific contacts with $\beta_2$AR (*Cherezov et al., 2007*; *Hanson et al., 2008*), suggesting that $\beta_2$AR has binding sites for cholesterol. Spectroscopic (*Gater et al., 2014*) and MD simulation (*Cang et al., 2013*; *Prasanna et al., 2014*; *Lee et al., 2012*) studies have reported direct interactions between cholesterol and GPCRs, including $\beta_2$AR. Experimental data show that cholesterol binding to $\beta_2$AR changes its structural properties (*Hanson et al., 2008*; *Zocher et al., 2012*). Cholesterol is also necessary in crystallizing $\beta_2$AR (*Cherezov et al., 2007*; *Hanson et al., 2008*), and cholesterol and its analogue cholesteryl hemisuccinate (CHS) have been exhibited to improve $\beta_2$AR stability (*Zocher et al., 2012*; *Loll, 2014*). Since the structure and function of GPCRs are closely related, cholesterol binding specifically to $\beta_2$AR is also expected to change the functional properties of the receptor. Indeed experimental studies indicate that cholesterol has a functional role in $\beta_2$AR

(*Paila et al., 2011*; *Pontier et al., 2008*; *Xiang et al., 2002*). Further, inhibition of β₂AR-associated signaling has been observed with increasing membrane cholesterol content (*Pontier et al., 2008*). However, as with GPCRs in general, the atomic-scale mechanism cholesterol uses to regulate β₂AR is not known. Does cholesterol modulate β₂AR activity through membrane-mediated effects by altering the physical properties of the membrane? Alternatively if regulation takes place through specific direct interactions, then what is the atom-scale mechanism?

We performed extensive atomistic MD simulations (totaling >100 µs, *Table 1*) to clarify the mechanism responsible for the modulatory role of cholesterol on β₂AR. In essence, we show that as cholesterol concentration reaches ~10 mol%, the conformational distribution of β₂AR is drastically altered. The mechanism of action is based on the binding of cholesterol at specific high-affinity sites of the receptor.

## Results

### Cholesterol restricts β₂AR conformation

We first studied the impact of cholesterol on the conformational distribution of β₂AR by systematically increasing the cholesterol concentration from 0 to 40 mol% in a DOPC (1,2-dioleoyl-*sn*-glycero-3-phosphocholine) bilayer. Crystallographic studies and previous biophysical and biochemical studies have shown that helices 5–6 (H5-H6) (*Figure 1A*) constitute a highly dynamic region of β₂AR (*Kobilka, 2013*). Upon activation, the most dramatic conformational change, which is conserved among many GPCRs, is a 7–14 Å outward movement of the intracellular end of H6 from the heptahelical core of the receptor (*Manglik and Kobilka, 2014*; *Kobilka, 2013*). The large rearrangement in the G protein-coupling interface is accompanied by a comparatively subtle change in the ligand-binding pocket. In a conformational change from the inactive to the active state β₂AR, H5 (around S207$^{5.46}$) has been found to move inward by 2 Å to establish an optimal interaction between the agonist and the two anchor sites (D113$^{3.32}$/N312$^{7.39}$ and S203$^{5.42}$/S204$^{5.43}$/S207$^{5.46}$) on the receptor (*Kobilka, 2013*).

In the present work where we started from the inactive structure of β₂AR (*Manna et al., 2015*), we calculated the distance between the Cα atoms of D113$^{3.32}$ and S207$^{5.46}$ (referred to as L$_L$) to measure the displacement of H5 in the ligand-binding site, and the distance between the Cα atoms of R131$^{3.50}$ and E268$^{6.30}$ (referred to as L$_G$) to determine the displacement of H6 in the G protein-binding site (*Figure 1A*); the position of H3 does not change noticeably (RMSD < 0.8 Å) during the simulations. These two parameters (L$_L$ and L$_G$) have been used in many previous studies to monitor changes in β₂AR conformation (*Manglik and Kobilka, 2014*; *Kobilka, 2013*; *Nygaard et al., 2013*; *Dror et al., 2009*; *Ozcan et al., 2013*; *Dror et al., 2011*; *Manna et al., 2015*), thus here we discuss the conformational distribution of the receptor as a function of L$_L$ and L$_G$ (*Figure 1B,C* and *Figure 1—figure supplement 1*). In the inactive crystal structure, the L$_L$ and L$_G$ values are 12.07 and 11 Å, respectively (*Hanson et al., 2008*).

In a cholesterol-free DOPC bilayer, we find β₂AR to adopt a wide range of conformations with L$_L$ varying between ~11.5–17.5 Å and L$_G$ ranging between ~7.5–12.5 Å (*Figure 1B*). The receptor populates two major conformational states. One of them has a relatively open G protein site (L$_G$ being 10–12 Å) and a smaller ligand-binding site (L$_L$ ~ 13 ± 1 Å). The other conformation is characterized by a shift of ~3–4 Å from the intracellular end of H6 towards the receptor core that blocks the G protein interface (L$_G$ ~ 8.5 Å). At the same time, the ligand-binding pocket expands as the extracellular part of H5 moves ~ 4–5 Å away from H3 (L$_L$ now ~16 ± 1 Å). This conformation represents an alternative inactive structure of the receptor, as both changes occur in the opposite direction compared to the case of agonist binding (*Kobilka, 2013*); we do not observe any transition to the active state of β₂AR. *Figure 1D* shows the receptor oscillating between the different inactive conformations during 2.5 µs. The closing of the intracellular G protein-binding surface is found to correlate with the opening of the extracellular ligand-binding pocket, and vice-versa (*Figure 1D*). The conformational correlation between the two distal sites supports the view of allosteric regulation in GPCRs (*Kobilka, 2013*; *Ozcan et al., 2013*).

In the presence of cholesterol, the picture changes quite dramatically. With a cholesterol concentration of 10 mol%, the conformational flexibility of β₂AR reduces significantly (*Figure 1C*). The receptor stays predominantly in one conformation and no further opening of the ligand-binding site

**Table 1.** Descriptions of systems simulated: $\beta_2$AR in bilayers with varying lipid compositions. 'Chol' stands for cholesterol.

| | Systems[*] | Initial lipid arrangement around $\beta_2$AR | Lipids | Sterol mol % | No. of repeats[†] | Time (µs)[‡] | |
|---|---|---|---|---|---|---|---|
| | DOPC | Random | DOPC | 0 | 3 | 3×2.5 | |
| | DOPC-active | Random | DOPC | 0 | 3 | 3×2.5 | |
| CHOL | Chol2 | Random | DOPC + Chol | 2 | 3 | 3×2.5 | RANDOM |
| | Chol5 | Random | DOPC + Chol | 5 | 3 | 3×2.5 | |
| | Chol10 | Random | DOPC + Chol | 10 | 3 | 3×2.5 | |
| | Chol25 | Random | DOPC + Chol | 25 | 2 | 2×2 | |
| | Chol40 | Random | DOPC + Chol | 40 | 3 | 3×2.5 | |
| | Chol40-active | Random | DOPC + Chol | 40 | 3 | 3×2.5 | |
| CHS | CHS10 | Random | DOPC + CHS | 10 | 2 | 2×2 | |
| | CHS40 | Random | DOPC + CHS | 40 | 2 | 2×2 | |
| | CHSA10 [A for anionic] | Random | DOPC + CHSA | 10 | 1 | 2 | |
| | CHSA40 | Random | DOPC + CHSA | 40 | 1 | 2 | |
| OXYSTEROL | 27-OH-Chol | Random [16 mol % Chol was randomly replaced by 27-OH-Chol] | DOPC + Chol + 27-OH-Chol | 25 (4 mol% 27-OH-Chol + 21 mol % Chol) | 3 | 2 + 1 + 1 | |
| | 4β-Chol | Random [16 mol% Chol was randomly replaced by 4β-OH-Chol] | DOPC + Chol + 4β-OH-Chol | 25 (4 mol% 4β-OH-Chol + 21 mol % Chol) | 3 | 1 + 1 + 1 | |
| | Chol-Bound[§] | 8 cholesterols bound at sites predicted by simulations | DOPC + Chol | 1.9 | 3 | 3×2.5 | BOUND |
| | Chol-IC1 | 2 Chol bound at IC1 | DOPC + Chol | <1 | 2 | 2×2 | |
| | CHS-IC1 | 2 CHS bound at IC1 | DOPC + CHS | <1 | 1 | 2 | |
| | CHSA-IC1 | 2 CHSA bound at IC1 | DOPC + CHSA | <1 | 1 | 2 | |
| | PC-20:0–22:1 c13 [Double bond at carbon 13] | Random | PC-20:0–22:1 c13 | 0 | 3 | 3×1.5 | |
| | Pyrene20 | Random | DOPC + 20 mol% pyrene | 0 | 3 | 3×1.5 | |

*In the DOPC-active and Chol40-active systems, we used the active-state conformation of the receptor as the starting structure; for all the other systems, we used the inactive conformation.

†For systems with no sterols initially bound to $\beta_2$AR, i.e., the systems which started with a random distribution of lipids, a number of different repeat simulations for each lipid composition were performed with different initial lipid arrangements around the receptor. For systems with sterols initially bound to $\beta_2$AR (see[d] and BOUND), different replicas were generated with different starting velocities.

‡Listed are the simulation times of production simulations; the equilibration time of the systems (100 ns) is not included.

§In the Chol-Bound system, eight cholesterol molecules were initially (at time zero of the simulation) bound at eight binding sites predicted by the present simulations, while the rest of the system had no cholesterol at all.


or the opening/closing of the G protein-binding site is observed, unlike in a cholesterol-free membrane. As shown in ***Figure 1E***, $L_L$ and $L_G$ fluctuate around ~13 and~9.5 Å, respectively. The slowing down of the movements of H5 and H6 correlates with the observed high-density spots of cholesterol at these helices (IC2 and EC1 in ***Figure 2*** discussed in detail below). To further quantify this, ***Figure 1F*** depicts the standard deviation for the fluctuations of the intracellular and extracellular ends of H5 and H6, when these ends are bound or unbound to cholesterol. The data show that the deviations of these helices from their respective average positions are much smaller when they are

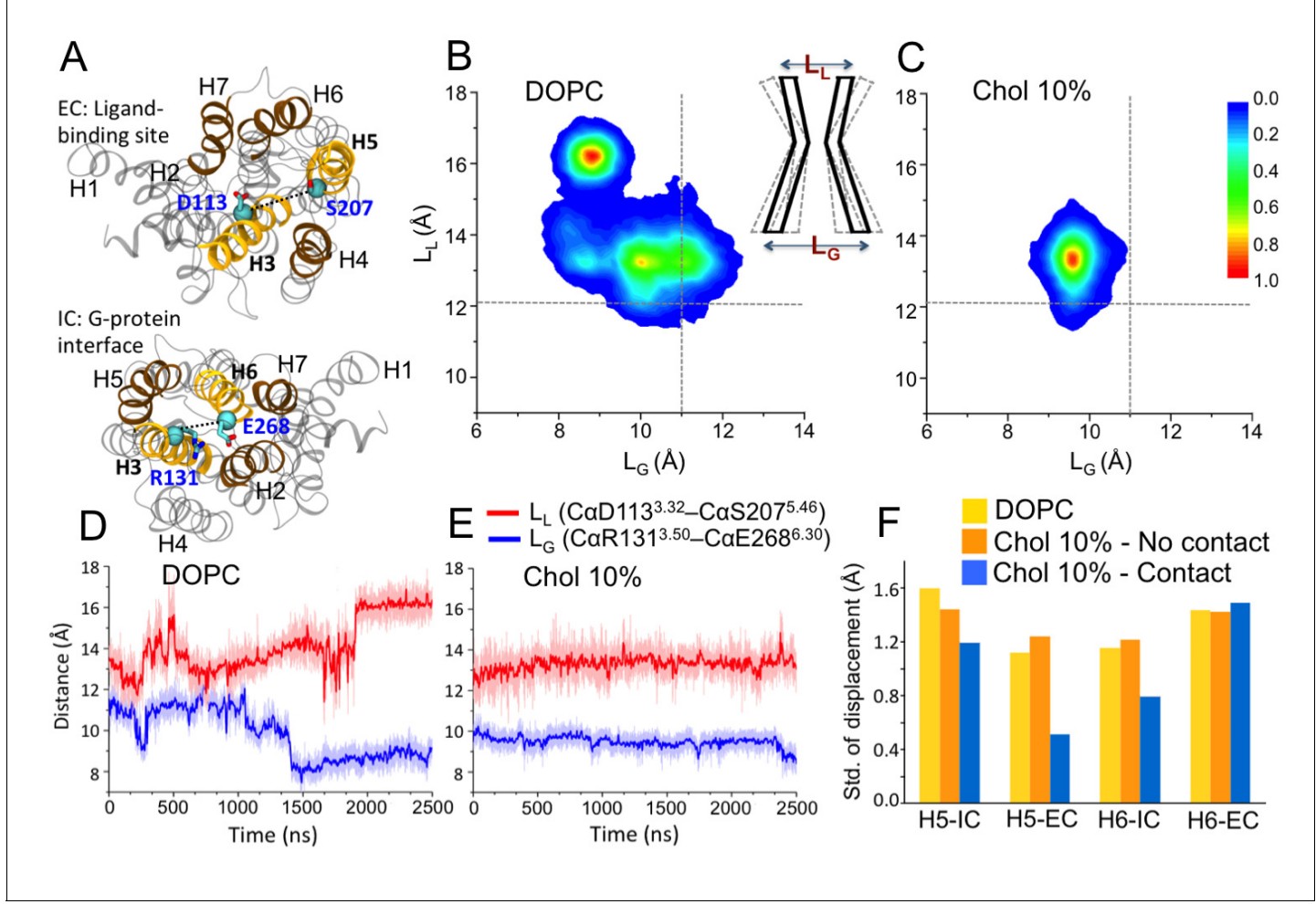

**Figure 1.** Conformational dynamics of $\beta_2$AR. (A) The distances between the C$\alpha$ atoms of D113$^{3.32}$–S207$^{5.46}$ (distance defined as $L_L$) and R131$^{3.50}$–E268$^{6.30}$ ($L_G$) pairs used to measure the fluctuations at the ligand and G-protein binding sites, respectively. (B–C) The conformational distributions of $\beta_2$AR in membranes with 0 and 10 mol% cholesterol (Chol) as a function of $L_L$ and $L_G$. The gray dotted lines represent the corresponding $L_L$ and $L_G$ values in the inactive crystal structure of $\beta_2$AR (*Hanson et al., 2008*). The cartoon diagram shows the fluctuations of $L_L$ and $L_G$ at the ligand and G-protein binding sites of the receptor, respectively. (D–E) The time evolution of $L_L$ (light red) and $L_G$ (light blue) in systems with 0 and 10 mol% cholesterol. Corresponding 50-point running averages are shown in dark colors. (F) Standard deviation for the distribution of the distance between the intracellular (IC) (or extracellular (EC)) end of H5 and its average position, and its dependence on whether the given end of H5 is in contact with cholesterol or not; similarly for H6.

The following figure supplement is available for figure 1:

**Figure supplement 1.** Conformational distributions of $\beta_2$AR in lipid bilayers with various cholesterol (Chol) concentrations.

bound to cholesterol. The effect is particularly strong for the extracellular end of H5 at the ligand-binding site and for the intracellular end of H6 at the G protein-binding site.

The restricted dynamics of $\beta_2$AR is also observed at higher cholesterol concentrations (25 and 40 mol%; *Figure 1—figure supplement 1D,E*). In these cases, the receptor samples a similar conformational space as observed with 10 mol% cholesterol. At lower concentrations (2 and 5 mol%), the distribution of the receptor's conformation is much wider (*Figure 1—figure supplement 1A,B*). Particularly when the membrane contains a very small percentage of cholesterol (2 mol%), the range of conformations accessible to $\beta_2$AR is almost comparable to that of a cholesterol-free membrane.

A broad conformational distribution (*Figure 1—figure supplement 1F*) is also observed in control simulations, where eight cholesterol molecules were initially placed at the cholesterol-binding sites

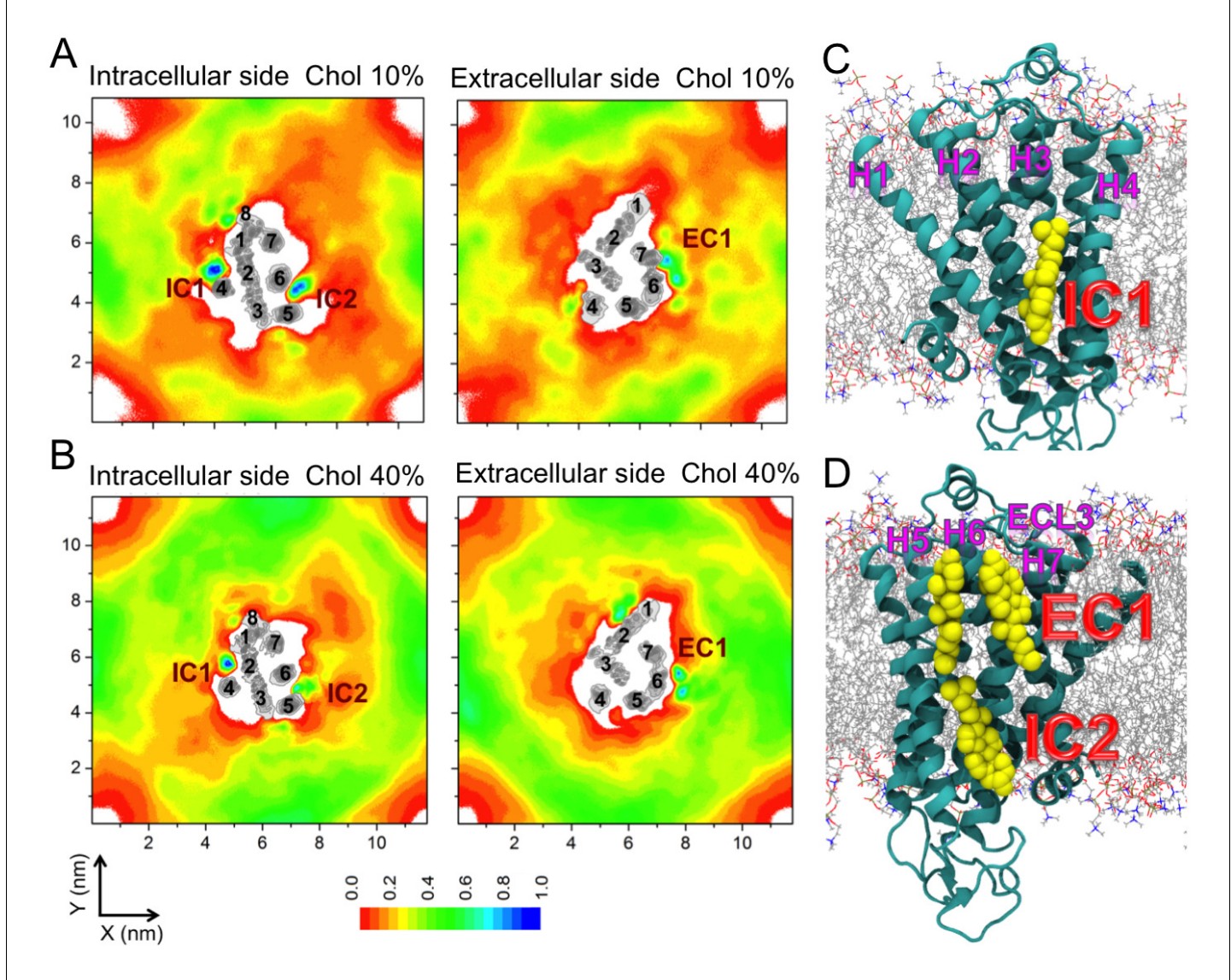

**Figure 2.** Cholesterol interaction sites on β2AR. (A–B) 2D number densities of cholesterol (Chol) around β2AR. The data are averaged over all independent trajectories for a given cholesterol concentration (*Table 1*) and normalized with respect to the maximum density for that particular cholesterol concentration. The intracellular (IC) and extracellular (EC) bilayer leaflets are depicted separately. The major cholesterol interaction sites (IC1, IC2 and EC1) are marked in the density plots. The IC and EC sides of the transmembrane regions (H1–H7) of β2AR are shown in gray scale (the darker the color, the higher is the number density) and numbered accordingly. (C–D) Cartoon representation of three main cholesterol interaction sites in β2AR. IC1 (H1–H4) and IC2 (H5–H6) are located on the intracellular side, and EC1 comprised of two closely placed cholesterols between H5-H6 and H6-ECL3-H7 is located on the extracellular side of β2AR.

The following figure supplements are available for figure 2:

**Figure supplement 1.** Residues of β2AR involved in cholesterol binding, and cholesterol interaction sites on β2AR.

**Figure supplement 2.** Sequence alignment of β2AR orthologues around the cholesterol-binding site IC1.

**Figure supplement 3.** Sequence alignment of β2AR orthologues around the cholesterol-binding site IC2.

**Figure supplement 4.** Sequence alignment of β2AR orthologues around the cholesterol-binding site EC1.

**Figure supplement 5.** Cholesterol density around the receptor at low cholesterol concentrations.

*Figure 2 continued*

**Figure supplement 6.** Structure of cholesterol analogues and properties of sterol-containing bilayers.
**Figure supplement 7.** Interactions of cholesterol and cholesterol-like molecules with $\beta_2$AR.
**Figure supplement 8.** Densities of sterols around $\beta_2$AR.
**Figure supplement 9.** Conformational distributions of $\beta_2$AR in lipid bilayers with different cholesterol analogues.
**Figure supplement 10.** IC1 interaction site.

of $\beta_2$AR predicted by our simulations (see below), and this receptor-cholesterol complex was then embedded in a cholesterol-free membrane. Here (*Figure 1—figure supplement 1F*) the concentration of cholesterol in the annular region is therefore high in the beginning of the simulation, while it is zero elsewhere. Cholesterols dissociate from $\beta_2$AR during the course of the simulation (discussed in detail below) and at long times the system corresponds to a dilute (cholesterol-poor) system, where the total average cholesterol concentration is low (1.9 mol%). One finds that as the data are averaged over the simulation period, the conformational behavior (*Figure 1—figure supplement 1F*) translates from cholesterol-rich (*Figure 1—figure supplement 1E*) to cholesterol-poor behavior (*Figure 1—figure supplement 1A,B*).

Further, we studied the effect of cholesterol on the active conformation of $\beta_2$AR in its apo form in the absence of the G protein (*Rasmussen et al., 2011*). In the active state, the intracellular end of H6 is splayed outward from the helical bundle, providing room for the G protein (*Figure 3A*). We observe inward swinging of H6 towards H3 in the absence of cholesterol (which occurred in two out of three replica simulations). As shown in *Figure 3B,E*, the intracellular end of H6 spontaneously approaches H3 with $L_G$ dropping from 18.97 Å in the starting active conformation to ~11.5 Å that is comparable to the crystallographically observed inactive conformation of $\beta_2$AR ($L_G \sim 11$ Å) (*Hanson et al., 2008*). Such spontaneous deactivation of the receptor in the absence of the intracellular binding partner and cholesterol is in agreement with recent simulations (*Dror et al., 2011*; *Neale et al., 2015*) and experimental studies (*Rosenbaum et al., 2011*). Meanwhile, with 40 mol% cholesterol, we observe that the active-like open conformation is stable during the simulations (*Figure 3—figure supplement 1*). As shown in *Figure 3C,E*, the $L_G$ value remains stable around 16.5 Å and no deactivation is observed unlike in cholesterol-free systems. Interestingly, here again we found a high cholesterol density at the intracellular segments of H5-H6 (IC2 in *Figure 3C,D*,F as discussed in detail below).

These results show that cholesterol restricts the intrinsic conformation dynamics of $\beta_2$AR and governs changes between different conformational states, thereby modulating its function.

## Specific binding of cholesterol

In all of the simulations (*Table 1*), cholesterol is observed to diffuse spontaneously to the receptor's surface. Time-averaged two-dimensional (2D) number density maps demonstrate that there are preferred cholesterol positions around $\beta_2$AR (*Figure 2A,B*).

Localized cholesterol hot spots are often used as an indicator of potential cholesterol binding sites. We identify three such cholesterol interaction sites – two on the intracellular side (IC1 and IC2) and one on the extracellular side (EC1) (*Figure 2*, *Figure 2—figure supplement 1A,B*). Here we call them high-affinity sites since they reproducibly exhibit high cholesterol densities (normalized number density above 0.7) at different cholesterol concentrations (*Figure 2A,B*) and also have large lifetimes as the below discussion shows. IC1 is a shallow groove formed by the intracellular parts of transmembrane helices H1-H4 and coincides well with the location of cholesterol observed in the crystal structure of $\beta_2$AR (*Cherezov et al., 2007*; *Hanson et al., 2008*). In IC2 cholesterol penetrates deep into the cleft between H5 and H6 on the intracellular side. A high density of cholesterol is observed at IC2 not only in the inactive but also in the active $\beta_2$AR conformation (*Figure 3C,D,F*), which suggests that this site is biologically important.

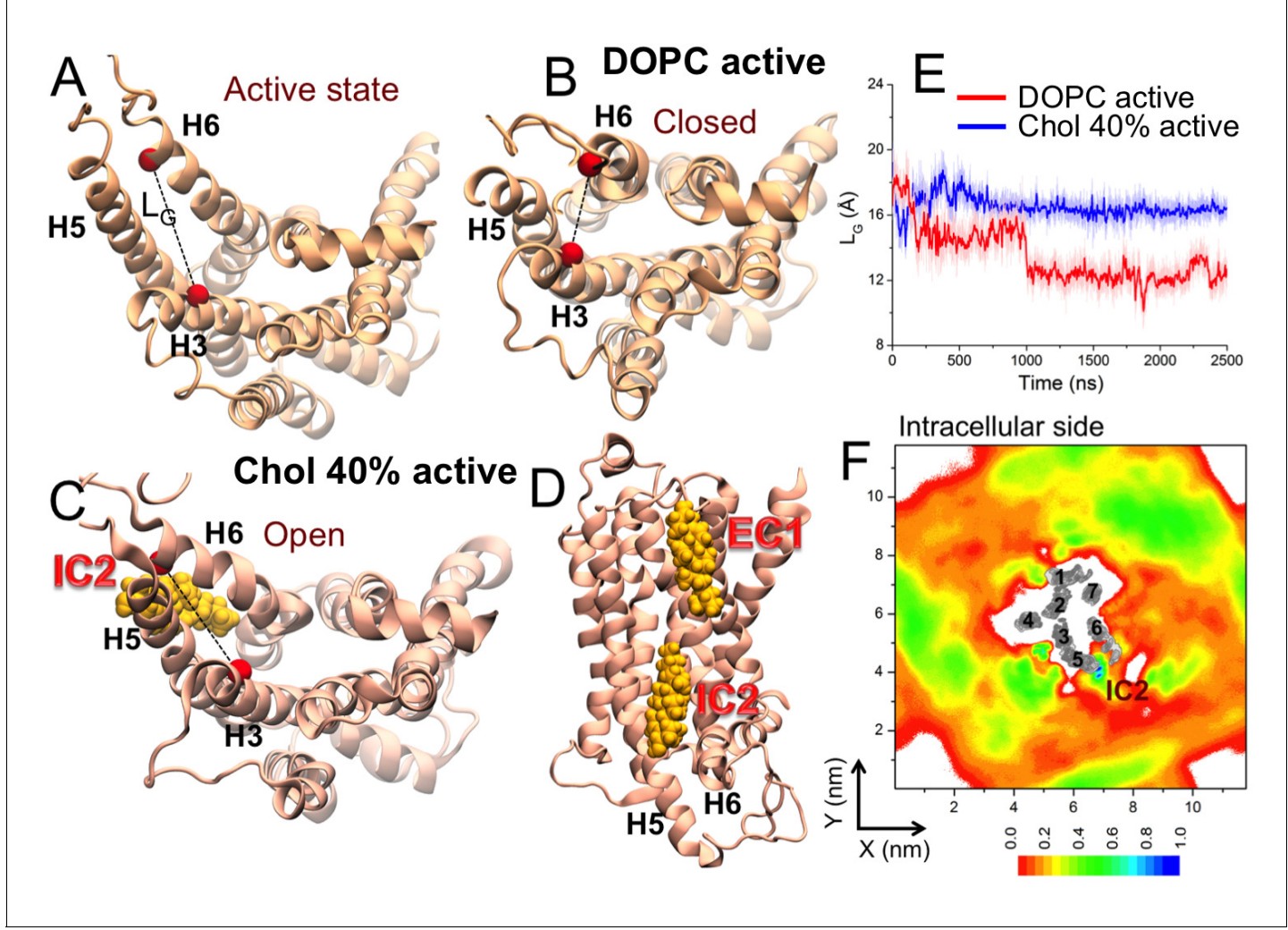

**Figure 3.** Effect of cholesterol on the active conformation of $\beta_2AR$. Cytosolic view of $\beta_2AR$ (**A**) in the beginning of a simulation (active state) as well as in representative simulation snapshots in (**B**) a DOPC bilayer and (**C**) in the presence of 40 mol% cholesterol. The dotted line represents the distance between the C$\alpha$ atoms of R131$^{3.50}$–E268$^{6.30}$ (defined as $L_G$), used to measure the fluctuation at the G protein-binding site. (**D**) Simulation snapshot (in the presence of 40 mol% cholesterol) showing cholesterol binding at the interaction sites of $\beta_2AR$. (**E**) The time evolution of $L_G$ in systems with 0 (light red) and 40 mol% cholesterol (light blue). Corresponding 50-point running averages are shown in dark colors (red, blue). (**F**) 2D number densities of cholesterol around $\beta_2AR$ (cytosolic view). The individual transmembrane helixes of $\beta_2AR$ are numbered and shown in gray scale (as in *Figure 2A,B*).

The following figure supplement is available for figure 3:

**Figure supplement 1.** Conformational distribution of $\beta_2AR$ starting from the active state.

EC1 is comprised of two closely spaced cholesterol hot spots located in the extracellular part of H5-H6 and H6-ECL3-H7 (where ECL stands for the extracellular loop). The occupancy of two cholesterol molecules at EC1 is in good agreement with the crystal structure of the adenosine receptor A$_{2A}$AR (*Liu et al., 2012*), while IC2 is so far unidentified among the experimentally determined structures (*Gater et al., 2014*). Notably, the cholesterol binding residues of the three interaction sites are conserved to a large degree among $\beta_2AR$ orthologues (*Figure 2—figure supplement 2*, *Figure 2—figure supplement 3*, *Figure 2—figure supplement 4*), indicating that these sites have conserved during the evolution of the receptor. In addition, a few comparatively low-affinity cholesterol binding sites (IC3-4, EC2-3) with 10 and 40 mol% cholesterol are observed (*Figure 2—figure supplement 1*). When cholesterol concentration is lowered below 10 mol%, many of the interaction sites, particularly IC1 and EC1, are occupied by cholesterol at concentrations as low as 5 mol% (*Figure 2—figure*

*supplement 5*). A few sites (IC2 and EC1) are visited, though transiently, by cholesterol even at 2 mol% (*Figure 2—figure supplement 5*).

In addition to the above-discussed cholesterol hot spots, we observed two sites with comparatively weak cholesterol occupancies (reproducible at both 10 and 40 mol% cholesterol concentrations): IC3 between H3 and H5, and IC4 between H1 and H8, both on the intracellular side (*Figure 2A,B* and *Figure 2—figure supplement 1*). IC4 recaptures the predicted cholesterol position at the dimerization interface of $\beta_2$AR found by X-ray crystallography (*Cherezov et al., 2007*). Besides these, another site with a low cholesterol density was observed near the extracellular part of H3-H4 (EC2) in the 10 mol% cholesterol system, and a high-density site was observed on the extracellular side of H1-H2-EC1 (EC3) in the 40 mol% cholesterol system (*Figure 2A,B*).

Concluding, we find cholesterol to bind to $\beta_2$AR in specific binding sites. These sites are in agreement with those found in the crystallographic structures of GPCRs (*Cherezov et al., 2007*; *Hanson et al., 2008*; *Gimpl, 2016*; *Warne et al., 2011*; *Liu et al., 2012*; *Gater et al., 2014*).

## Membrane-mediated interactions not the key

Is it possible that the effects we observed on $\beta_2$AR conformation could be due to cholesterol-induced changes in membrane properties, and the changes in $\beta_2$AR would hence not be due to the specific direct binding of cholesterol in the hot spots? To unlock this issue, we study the conformational properties of $\beta_2$AR in cholesterol-free membranes whose physical properties (thickness, order, diffusion) match those of membranes with a large concentration of cholesterol.

A. *Effect of increased bilayer thickness.* We studied $\beta_2$AR embedded in a bilayer composed of long-chain mono-unsaturated phosphatidylcholine (PC) lipids PC-20:0/22:1 c13 (*Koynova and Caffrey, 1998*). The thickness of this membrane is larger than the thickness of a DOPC bilayer with 40 mol% cholesterol, while its lipid chain order is comparable to a DOPC bilayer with 5% cholesterol (*Figure 4—figure supplement 1A,B*). *Figure 4A* depicts that the increased bilayer thickness is unable to restrict the conformational dynamics of $\beta_2$AR. The receptor just adjusts itself to the hydrophobic mismatch by inducing bilayer thinning (4–8 Å) in its vicinity (*Figure 4B*).

B. *Effect of increased bilayer order.* We then studied $\beta_2$AR placed in a DOPC bilayer with 20 mol% pyrene, which is known to induce similar (ordering and condensing) effects as cholesterol (*Curdová et al., 2007*). *Figure 4D* highlights that pyrene does not show any preference for specific binding on the $\beta_2$AR surface except for the slowed-down diffusion of pyrene near the receptor surface. $\beta_2$AR exhibits a very broad conformational distribution, with $L_L$ and $L_G$ fluctuating between ~9–17.5 and ~7–13.5 Å, respectively (*Figure 4C*). This conformational behavior of the receptor is distinctly different from the one induced by $\geq$10 mol% cholesterol, although the order of the pyrene-containing bilayer is similar to a DOPC bilayer with 10 mol% of cholesterol (*Figure 4—figure supplement 1D*).

Summarizing, the changes in physical membrane properties, similar to those induced by cholesterol, do not restrict the conformational dynamics of $\beta_2$AR. We conclude that the cause of the observed changes in $\beta_2$AR conformation and dynamics is the specific binding of cholesterol to $\beta_2$AR.

## Binding lifetime depends on cholesterol

When cholesterol is specifically bound to $\beta_2$AR, how stable is the binding? *Figure 5* depicts the time-correlation function of cholesterol binding in the three main binding sites (IC1, IC2, EC1) on $\beta_2$AR and shows that at low cholesterol concentrations (2–5 mol%) the binding lifetime is short, of the order of 100 ns or less. However, at ~10 mol% there is a clear transition to longer lifetimes (see *Video 1* and *Video 2*) given that the lifetime of binding increases to the microsecond time scale for 10 and 40 mol% cholesterol.

In three control simulations where cholesterols were initially bound at the eight cholesterol-binding sites identified in our simulations and no further cholesterol was in the bilayer (*Figure 5—figure supplement 1*), cholesterols underwent rapid unbinding from the majority of the binding sites in a timescale of tens to hundreds of nanoseconds (*Figure 5—figure supplement 1*), similarly to the short binding lifetime observed for cholesterol-poor systems (2 mol%, *Figure 5*). However, at a few sites cholesterol stayed for the entire simulation time (IC1 and IC2 in two out of three simulations) or dissociated in the µs timescale (IC3 and EC3 in one simulation).

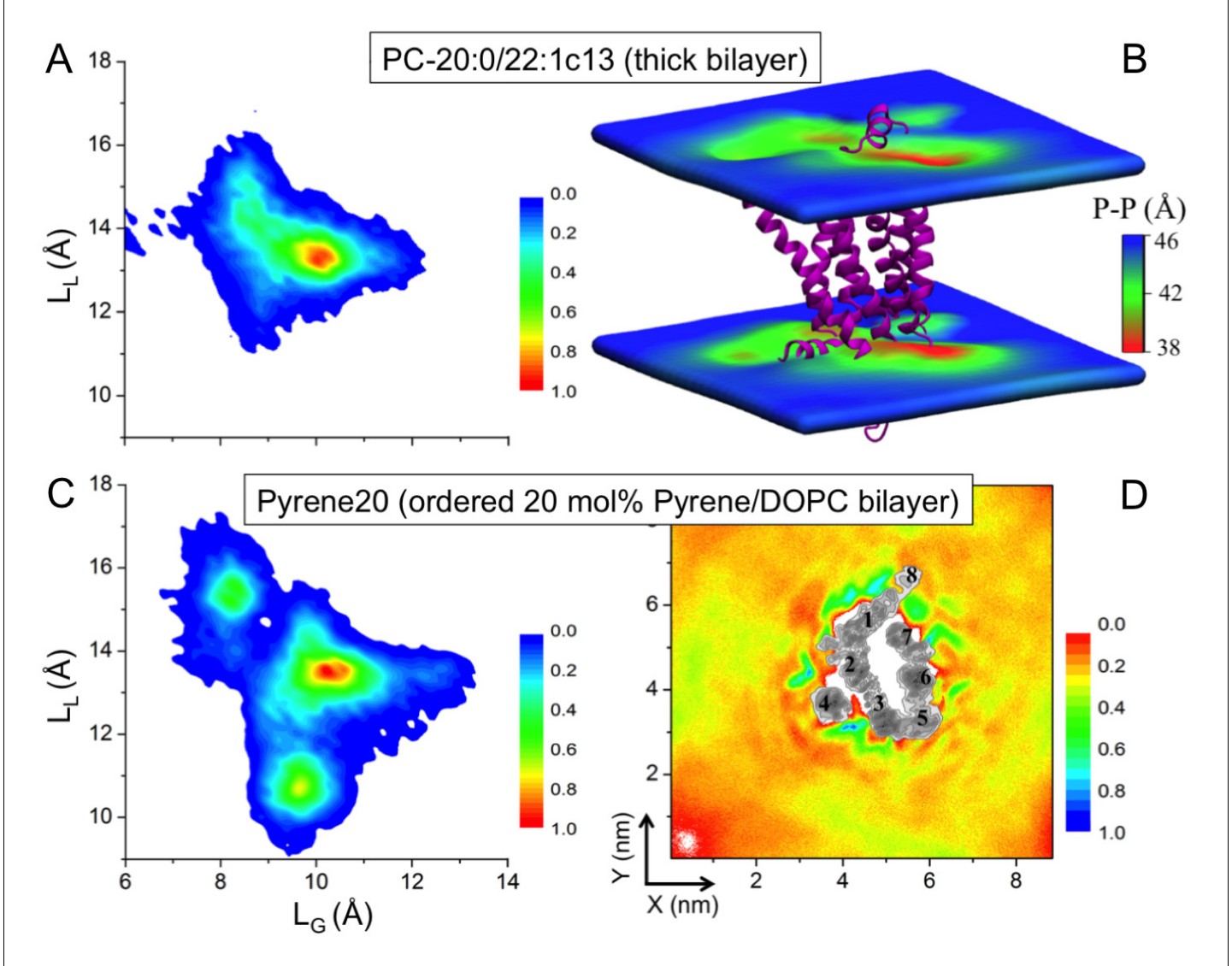

**Figure 4.** Impact of membrane-mediated effects on the β2AR conformation. The conformational distribution of β2AR in bilayers composed of (**A**) long-chain PC-20:0/22:1 c13 lipids and (**C**) DOPC with 20 mol% pyrene (Pyrene20). (**B**) 3D-distribution of bilayer thickness in the thicker PC-20:0/22:1 c13 membrane. The receptor is shown as a purple cartoon. (**D**) 2D number density of pyrene around β2AR.

The following figure supplement is available for figure 4:

**Figure supplement 1.** Properties of thick and/or ordered cholesterol-free bilayers.

The results show that the lifetime of cholesterol is of the order of microseconds in the high-affinity binding sites, where the lifetime at large cholesterol concentrations is largely independent of cholesterol concentration.

## Cholesterol analogues interact with β2AR

We next explore how cholesterol analogues, in comparison to cholesterol, interact with β2AR. We focus on four different analogues (*Table 1*): (i-ii) cholesteryl hemisuccinate (CHS) and its deprotonated form (CHSA), and (iii-iv) two oxysterols, 4β-hydroxycholesterol (4β-OH-Chol) and 27-hydroxy-cholesterol (27-OH-Chol), oxidized at the cholesterol ring and tail, respectively (*Figure 2—figure supplement 6A*). As compared to cholesterol, CHS is a more water-soluble cholesterol ester and is

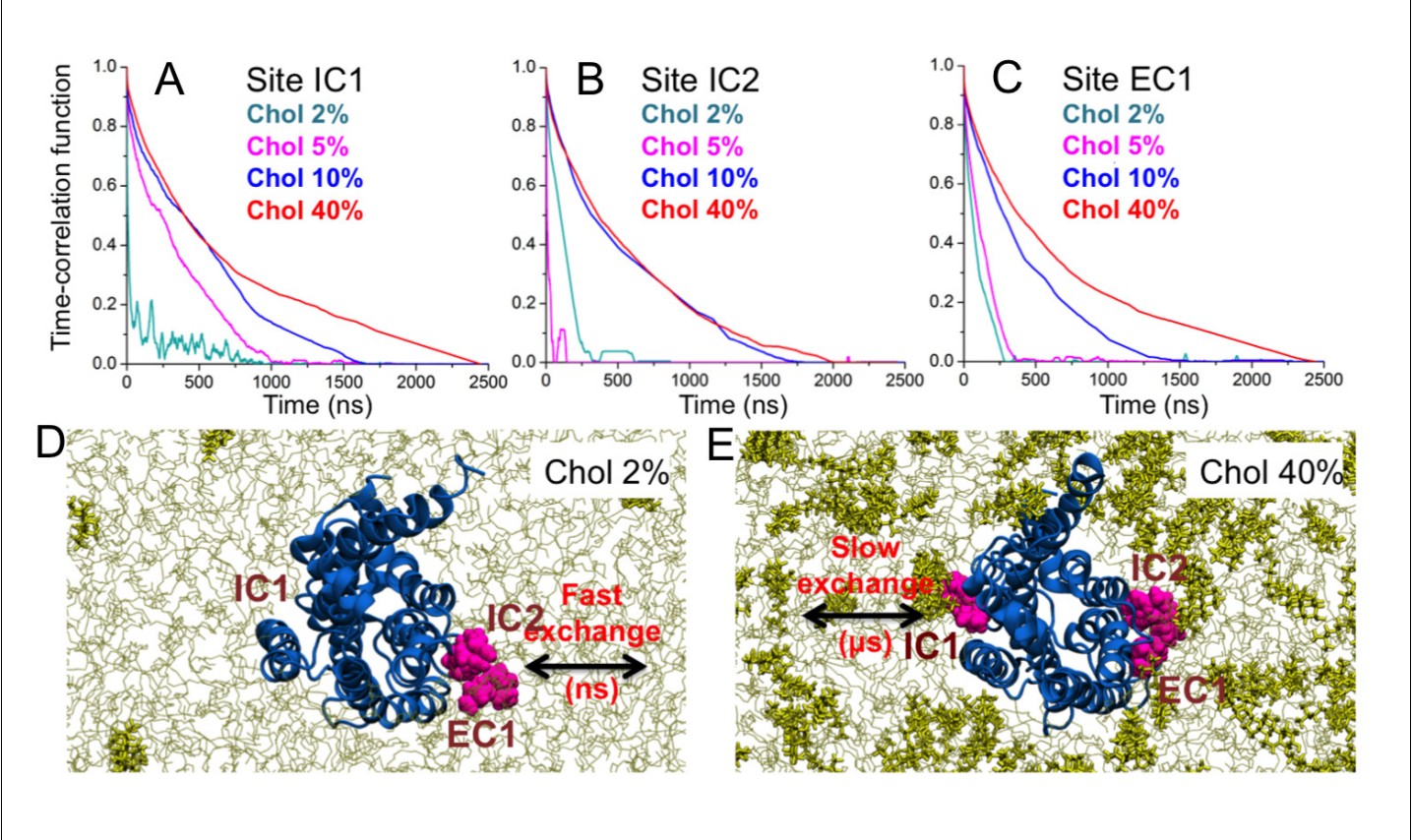

**Figure 5.** Binding time of cholesterol. (**A–C**) Time-correlation function of cholesterol (Chol) at the three major interaction sites (IC1, IC2, EC1) on the β₂AR surface. Initially cholesterol is bound to the site (distance ≤ 0.5 nm) and the correlation function describes the probability that cholesterol remains bound to the given site for increasing time. Data are shown for DOPC-cholesterol membranes with 2, 5, 10, and 40 mol% of cholesterol. (**D–E**) Schematic representation showing the transition from fast to slow exchange as cholesterol concentration increases from 2 to 40 mol%. Color code: β₂AR (blue), DOPC (thin grey lines), cholesterol molecules bound to the interaction sites (purple), and other cholesterol molecules not bound to the receptor (yellow sticks).

The following figure supplement is available for figure 5:

**Figure supplement 1.** Interaction of cholesterol with β₂AR.

widely used in structural biology and biophysical studies as a cholesterol analogue (*Zocher et al., 2012*; *Loll, 2014*). Oxysterols, on the other hand, are derivatives of cholesterol with additional oxygen-containing substitutions at different positions of cholesterol (*Olkkonen and Hynynen, 2009*; *Kulig et al., 2015a*; *Neuvonen et al., 2014*). Due to the structural similarities with cholesterol, these analogues mimic cholesterol as to the effects on membrane properties (e.g., increasing bilayer order and thickness), although to different extents (*Figure 2—figure supplement 6*) (*Kulig et al., 2015a*, *2015b*).

CHSA is found to interact strongly with β₂AR due to the enhanced electrostatic coupling resulting from its negatively charged head-group (*Figure 2—figure supplement 7*), however it favors to reside around the receptor at locations different from those of cholesterol (*Figure 2—figure supplement 8A,B*). Meanwhile, CHS closely mimics the behavior of cholesterol (*Figure 2—figure supplement 7*). Among the three major cholesterol interaction sites observed in our simulations, we find a very high CHS density at IC2 (*Figure 2—figure supplement 8C–F*). High occupancy of CHS is also observed near IC1 (at 40 mol% CHS concentration) but not at all at EC1. Occupancy of CHS at IC1 is consistent with the crystal structure of β₁AR (*Warne et al., 2011*).

4β-OH-Chol interacts only weakly with β₂AR (*Figure 2—figure supplement 7*). Almost all of the interaction sites on the receptor surface are occupied by cholesterol rather than 4β-OH-Chol

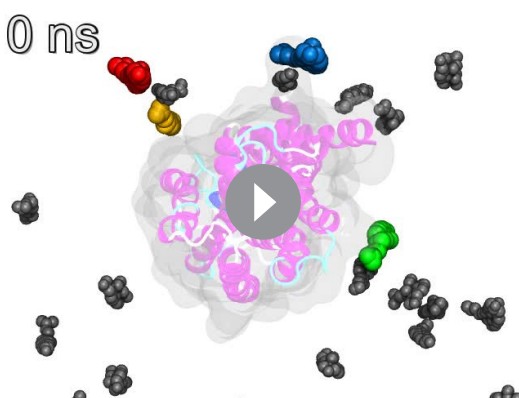

**Video 1.** Spontaneous binding/unbinding of cholesterol at the three main cholesterol interaction sites of β₂AR during a 2.5-μs simulation with 10 mol% of cholesterol. Cholesterols interacting at the cholesterol-binding sites are highlighted (yellow at IC1; green at IC2; and blue and red at EC1). Other cholesterols are shown in gray. For clarity, other lipids in a membrane are not shown.

(*Figure 2—figure supplement 8G–J*). As a result, the average density maps, showing the lateral arrangement of these sterols around β₂AR, are similar to those of 10 and 40 mol% cholesterol systems (*Figure 2A*), and reproducible. Unlike 4β-OH-Chol, 27-OH-Chol prefers to interact with the receptor directly (*Figure 2—figure supplement 7*). For the IC1 site, 27-OH-Chol competes, though weakly, with cholesterol, while at EC1 and IC3, 27-OH-Chol exhibits preference over cholesterol (*Figure 2—figure supplement 8K–N*).

Altogether, our results show that also other cholesterol-like molecules interact with β₂AR and may occupy the same binding sites on the receptor surface as cholesterol. However, the effects of cholesterol-analogues on β₂AR are weaker compared to those induced by cholesterol (*Figure 2—figure supplement 9*). All the cholesterol analogues studied here have a rigid ring structure, yet their slightly different chemical compositions influence their occupancy as well as the strength of binding to the cholesterol-binding sites (*Table 2*). This is assessed here in terms of the van der Waals energy, which as a short-range interaction reflects how strongly two molecules are in contact and therefore serves as an appropriate measure for the gravity of lipid-protein binding in the binding site.

The results in *Table 2* show that among the three major interaction sites, the binding of CHS at IC1 is much weaker than that of cholesterol. At IC2 the strength of interaction of CHS and cholesterol is comparable. Meanwhile, the extracellular EC1 site remains unoccupied by CHS indicating the binding energy to be low. As to the two oxysterols, 4β-OH-Chol interacts with β₂AR only at EC1 and

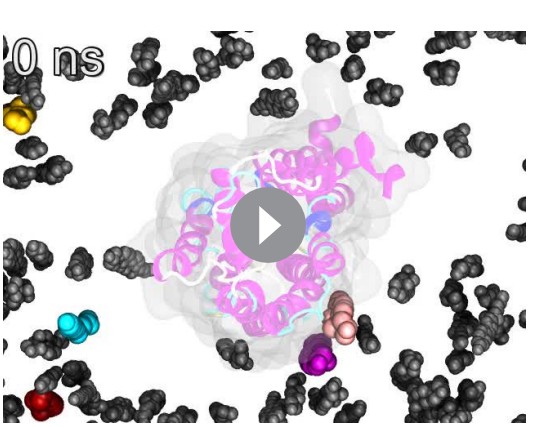

**Video 2.** Spontaneous binding/unbinding of cholesterol at the three main cholesterol interaction sites of β₂AR during a 2.5-μs simulation with 40 mol% of cholesterol. Cholesterols interacting at the cholesterol-binding interaction sites are highlighted (yellow and green at IC1; red, blue and orange at IC2; and pink, purple and cyan at EC1). Other cholesterols are shown in gray. For clarity, other lipids in a membrane are not shown.

the interaction is weak, while 27-OH-Chol binds at EC1 as tightly as cholesterol, but its interaction at the two other binding sites (IC1 and IC2) is much weaker than in the case of cholesterol. Concluding, CHS interacts at IC2 as strongly as cholesterol but its interactions at IC1 and EC1 are negligible compared to those of cholesterol. The oxysterol 27-OH-Chol interacts at EC1 as strongly as cholesterol but its interactions at IC1 and IC2 are negligible compared to those of cholesterol. The oxysterol 4β-OH-Chol does not interact with β₂AR to a significant degree.

These data can be considered in the context of molecular structures. In CHS, the difference compared to cholesterol is the additional chain bridged to the cholesterol structure via an ester bond (*Figure 2—figure supplement 6A*). This additional chain does not interfere binding at IC2, but it does alter the binding at IC1 and EC1. In 27-OH-Chol, the oxidation has taken place in the short acyl chain that is the terminal subunit of the molecule. This does not interfere the binding at EC1 but does alter the binding at IC1 and IC2. Finally, in 4β-OH-Chol, the oxidation has occurred in the rigid steroid moiety,

**Table 2.** Interactions* of sterols at the three high-affinity cholesterol-binding sites.

| Cholesterol/Cholesterol analogue | High-affinity cholesterol interaction sites | | | | | |
| | IC1 | | IC2 | | EC1 | |
| | vdW interaction energy (kJ/mol) | No. of contacts | vdW interaction energy (kJ/mol) | No. of contacts | vdW interaction energy (kJ/mol) | No. of contacts |
|---|---|---|---|---|---|---|
| Cholesterol† | −138.04 ± 0.20 | 141.02 ± 0.22 | −95.06 ± 0.12 | 90.65 ± 0.16 | −129.51 ± 0.29 | 104.38 ± 0.28 |
| CHS | −29.63 ± 0.14 | 28.78 ± 0.16 | −98.75 ± 0.11 | 96.30 ± 0.16 | - | - |
| 27-OH-Chol | −32.17 ± 0.30 | 34.95 ± 0.33 | −22.69 ± 0.23 | 28.41 ± 0.28 | −132.85 ± 0.27 | 120.20 ± 0.30 |
| 4β-OH-Chol | - | - | - | - | −41.80 ± 0.48 | 33.41 ± 0.42 |

* Shown are the total van der Waals (vdW) interaction energy and the number of contacts between cholesterol and $\beta_2$AR, when cholesterol is in the IC1, IC2, or EC1 binding site (and similarly for the cholesterol analogues).

† Calculations are based on systems having ≥10 mol% cholesterol. Shown here are the average values over different trajectories.

making the α-side of the molecule rougher. In cholesterol, the α-side is exceptionally flat. Given this change in surface roughness, and the importance of the surface-surface contact in lipid-$\beta_2$AR binding in the binding site, it is quite obvious why this oxysterol does not bind to any of the cholesterol binding sites (IC1, IC2, EC1).

The results support the view that the restriction of $\beta_2$AR dynamics arises from specific lipid binding to the receptor binding sites: the tighter the binding, the more is the receptor dynamics suppressed, and cholesterol induces the strongest effect.

## Discussion

Our results show that cholesterol has a preference to bind to $\beta_2$AR at specific locations on its surface. We identified three high-affinity cholesterol interaction sites in $\beta_2$AR (*Figure 2C,D*): IC1 (at the cleft of H1-H4 on the intracellular side), IC2 (H5-H6 on the intracellular side), and EC1 (the H5-H6-ECL3-H7 region on the extracellular side). IC1 and EC1 are in agreement with the locations of cholesterol found in GPCR crystal structures (*Cherezov et al., 2007*; *Hanson et al., 2008*; *Liu et al., 2012*). IC1 contains a cholesterol consensus motif that predicts cholesterol binding for 44% of human class A receptors (*Hanson et al., 2008*). Moreover, these binding sites appear to be evolutionarily conserved in $\beta_2$AR, which suggests their possible allosteric role in receptor function. A recent simulation study reported a correlation between cholesterol occupancy at IC1 and $\beta_2$AR dimerization (*Prasanna et al., 2014*). However, not much is known about the functional relevance of cholesterol binding to the other sites of $\beta_2$AR.

The present work for the inactive conformation of $\beta_2$AR shows that cholesterol binding at IC2 and EC1 (*Figure 2C,D*) strongly influences the conformational dynamics of $\beta_2$AR (*Figure 1*). In a cholesterol-free membrane the receptor samples multiple conformational states (*Figure 1B*) accounting for the high basal activity of $\beta_2$AR (*Manglik and Kobilka, 2014*; *Kobilka, 2013*). Our results show that the presence of cholesterol in high densities around H5-H6-H7 impedes the dynamic nature of the receptor. In cholesterol-containing (≥10 mol% cholesterol) membranes (*Figure 1C* and *Figure 1—figure supplement 1D,E*), the overall structural flexibility of the receptor is significantly reduced to one predominant conformation. We observed that in the presence of strongly bound cholesterol, H5 and H6 undergo much smaller displacements from their average positions as compared to the situation without cholesterol (*Figure 1F*). Cholesterol analogues that occupy the same interaction sites also restrict the $\beta_2$AR conformation (*Figure 2—figure supplement 9*), although their effects are weaker compared to those of cholesterol. Cholesterol or cholesterol-like molecules bound at these inter-helical clefts can thus confine the movement of the respective helices to a substantial degree, thus dampening the overall conformational dynamics of the receptor. At IC2 of inactive $\beta_2$AR, cholesterol pushes the intracellular end of H6 more towards the core of the helical bundle and prevents the outward movement of H6 required for G protein binding. The restriction of H6 movement by cholesterol is a potentially important allosteric effect, which can be used to modulate the receptor activity.

Interestingly, our study on the active-state β2AR also exhibits a high cholesterol density at IC2 (*Figure 3D,F*). Here cholesterol bound at IC2 acts as a spacer between H5-H6 and restricts the movement of H6, thereby stabilizing the open active-like conformation of the receptor (*Figure 3D*), while in the absence of cholesterol the receptor is more prone to undergoing spontaneous deactivation (*Figure 3E*; *Figure 3—figure supplement 1*). This result supports the postulate that cholesterol restricts the conformational dynamics of the receptor by binding at specific interaction sites and governs changes between different receptor states, therefore modulating its function. Moreover, cholesterol binding at IC2 in both inactive and active states of β2AR as found in our simulations highlights the biological relevance of this interaction site in allosteric regulation of the receptor conformation.

The highly conserved IC1 site shows no major influence on the mobility of H5-H6. On the other hand, IC1 exerts a stabilizing effect on H4 (*Figure 2—figure supplement 10*), in agreement with experiments (*Hanson et al., 2008*). As H4 is one of the weakest points of the β2AR fold, its decreased mobility may account for the enhanced stability of the receptor.

Cholesterol modulates the physical properties of membranes by increasing the bilayer thickness and order, and slowing down the dynamics. These general membrane effects can also influence the dynamic nature of a membrane protein (*Manna and Mukhopadhyay, 2011*). However, here we found that membrane-mediated interactions do not affect β2AR conformation to a significant degree (*Figure 4*).

GPCRs are signaling machines that function by toggling between multiple conformers (*Latorraca et al., 2016*). The dynamic nature of GPCRs has made their crystallization process extremely challenging (*Kobilka, 2013*). Besides techniques like protein engineering and use of detergents to increase the intrinsic stability of the receptor (*Loll, 2014*), cholesterol/CHS has emerged as a necessary component for crystallization of many GPCRs, including β2AR (*Cherezov et al., 2007*; *Hanson et al., 2008*; *Zocher et al., 2012*; *Loll, 2014*). Our work shows that in the presence of more than ~10 mol% cholesterol, inactive β2AR partly loses conformational variability and populates just one major conformation. Achieving conformational homogeneity is the key to crystallize membrane proteins (*Loll, 2014*). In agreement with our results, a recent experimental study showed that CHS impacts the conformational dynamics of a GPCR leading to a restricted conformational space (*Casiraghi et al., 2016*). Earlier it was experimentally reported that cholesterol induces a more compact conformational state of the oxytocin receptor (*Muth et al., 2011*). Our results are also in agreement with a recent dynamic single-molecule force spectroscopic study, which showed that CHS strengthens interactions that stabilize the structural segments in β2AR and thereby considerably increase the kinetic, energetic, as well as the mechanical stability of the receptor (*Zocher et al., 2012*). In addition, the function of adrenergic receptors is known to be modulated by cholesterol: cholesterol depletion enhances β2AR-associated signaling, while increased cholesterol content inhibits signaling (*Paila et al., 2011*; *Pontier et al., 2008*).

To our knowledge, the results presented in this work provide the first atomic-scale picture of how lipids can govern the conformation of membrane receptors through direct lipid-protein interactions in specific lipid binding sites, and hence dictate the state of a receptor. The receptor-cholesterol interactions, such as those observed in our simulations for β2AR, can conceivably govern the signaling of many GPCRs in the given protein family.

## Materials and methods

We performed all-atom molecular dynamics simulations of β2AR embedded in lipid bilayers with various lipid compositions (*Table 1*) using the GROMACS 4.6.x software package.

### Force field parameters

All simulations were performed using the GROMACS 4.6.x package (*Berendsen et al., 1995*; *Hess et al., 2008*). The all-atom OPLS-AA (optimized potentials for liquid simulations) force field was used to parameterize the protein, ions, and pyrene (*Jorgensen et al., 1996*; *Kaminski et al., 2001*). Force field parameters for cholesterol, cholesteryl hemisuccinate, and oxysterols were taken from previously published papers (*Manna et al., 2015*; *Kulig et al., 2015a*, *2015b*, *2014*). For the studied phosphatidylcholines (DOPC and PC-20:0/22:1 c13), we used new torsional and Lennard-Jones parameters derived for saturated (*Maciejewski et al., 2014*) and unsaturated hydrocarbons (*Kulig et al., 2015c*, *2016*) and the torsional potential developed for the glycerol backbone and the

phosphatidylcholine head group (*Maciejewski et al., 2014*). The TIP3P model, which is compatible with the OPLS parameterization, was used for water molecules (*Jorgensen et al., 1983*).

## Simulation protocols

All simulations of the systems considered in this work (*Table 1*) were performed under the isobaric-isothermal (NpT) ensemble. A time step of 2 fs was used for integrating the equations of motion. Periodic boundary conditions were applied in all three directions of the system. The temperature of the system was maintained at 310 K by employing the v-rescale (stochastic velocity rescaling) thermostat (*Bussi et al., 2007*) with a time constant of 0.1 ps. The temperatures of the receptor, lipids, and solvent molecules were controlled independently. The pressure of the system (1 bar) was maintained semi-isotropically using the Parrinello–Rahman barostat (*Parrinello and Rahman, 1981*) with a 1 ps time constant. The LINCS algorithm was applied to preserve hydrogen covalent bond lengths (*Hess et al., 1997*). Lennard-Jones interactions were cutoff at 1.0 nm. The particle mesh Ewald (PME) method (*Essmann et al., 1995*) was employed for long-range electrostatic interactions using a real space cutoff of 1.0 nm, β-spline interpolation (order of 6), and a direct sum tolerance of $10^{-6}$.

## Protein structure

The initial coordinates of $\beta_2$AR were taken from our recently published work (*Manna et al., 2015*), in which the structural modifications made for crystallization of the inactive $\beta_2$AR structure [PDB id: 3D4S] (*Hanson et al., 2008*) were reverted back to its original sequence. This inactive crystal structure of $\beta_2$AR bound to the partially inverse agonist timolol was heavily engineered to facilitate crystallization (*Hanson et al., 2008*). We reverted all the structural modifications from the experimentally determined structure, i.e., we removed mutations (E122$^{3.41}$W on the transmembrane helix H3 and the N187$^{5.26}$E mutation on the extracellular loop 2), removed the T4-lysozyme attached between the transmembrane helices 5 and 6, and replaced it with the missing intracellular loop 3. We did not attempt to model the unresolved N-terminal (32 residues) and C-terminal (71 residues) parts. The details of the procedure used to prepare the receptor model for our simulations are described elsewhere (*Manna et al., 2015*). In the present work, we considered the apo-receptor (without a ligand), as we were interested in the intrinsic dynamics of $\beta_2$AR.

For simulations with the active-state $\beta_2$AR conformation, the starting structure was taken from the crystal structure of the receptor bound to an agonist and a Gs protein (*Rasmussen et al., 2011*). Here again we considered the apo-form of the receptor without the ligand and the G protein. Additionally, we removed the lysozyme and modeled the missing loop regions (A176-H178 and F240-F264), but the mutations were kept as such.

## System setup

We simulated $\beta_2$AR embedded in a number of lipid bilayers (*Table 1*) with varying lipid composition. The lipid contents used in the studies were as follows:

- DOPC bilayers with different cholesterol (Chol) concentrations: 0, 2, 5, 10, 25, and 40 mol%.
- DOPC bilayers with a cholesterol analogue cholesteryl hemisuccinate (CHS; 10 and 40 mol%) or its deprotonated form CHSA (10 and 40 mol%). CHS is known to enhance the stability of GPCRs. It is frequently used for GPCR characterization (*Zocher et al., 2012*; *Yao and Kobilka, 2005*).
- DOPC bilayers mixed with several sterols: 21 mol% cholesterol and 4 mol% oxidized sterol (4β-hydroxy-cholesterol (4β-OH-Chol) or 27-hydroxy-cholesterol (27-OH-Chol)). Oxysterols used in this study are among the most common oxysterols found in human serum (*Olkkonen and Hynynen, 2009*; *Kulig et al., 2015a*).
- A single-component bilayer composed of the long-tail monounsaturated phospholipid PC-20:0/22:1 c13.
- DOPC bilayers with 20 mol% pyrene.

The lipid bilayers (without $\beta_2$AR) were constructed using in-house scripts, and they were subsequently solvated with water. These lipid bilayers were then equilibrated for 100–200 ns.

Next, $\beta_2$AR was placed into the above-mentioned pre-equilibrated bilayers in such a manner that the lipid arrangement around the receptor was completely random and that there was no cholesterol or cholesterol analogue initially bound to $\beta_2$AR. For incorporating the receptor into a pre-

equilibrated lipid bilayer, we followed our recently published method, where the receptor was pushed into a lipid membrane from its side by applying a high lateral pressure on the system (*Javanainen, 2014*).

Each system contained one β₂AR and 256–512 lipids. Each of the systems was explicitly solvated by water. In all cases, counterions (8 Cl⁻ ions for β₂AR, and additional Na⁺ counter ions for bilayers containing the anionic CHSA) were added to maintain electroneutrality of the systems. NaCl salt was added to achieve the physiological salt concentration of 150 mM. Subsequently each system was energy minimized and then equilibrated in two stages with position restraints first on protein heavy atoms and then on the backbone. Following equilibration (100 ns), all restraints were released and the equilibrated systems were subjected to microsecond length (1–2.5 μs) production simulations. Multiple independent simulations were performed for each lipid composition, either by starting from a different lipid arrangement around β₂AR (for systems with no sterols initially bound to the receptor) or starting with different initial velocities (for systems with sterols initially bound to the receptor).

Additional simulations were performed where cholesterol or its analogues were initially attached to certain locations on the surface of the receptor, and this receptor-lipid complex was then embedded to a cholesterol-free DOPC bilayer. Here we performed two sets of control simulations. In one set of simulations, two cholesterol or CHS (neutral or anionic) molecules were bound at the cleft formed by the intracellular side of the transmembrane helices H1-4 as predicted from the crystal structure (*Hanson et al., 2008*). In another set of control simulations, cholesterol molecules were initially bound at the eight interaction sites of β₂AR predicted by our simulations (see discussion in the main article). The simulation conditions (as to counterions and salt, release of restrains, simulation times, etc.) were as described above.

The systems investigated in this study are summarized in *Table 1*. The total simulation time for the atomistic systems studied in this work covers a period of more than 100 μs.

## Analysis of helix deviation

For calculation of deviations of helix ends, we first calculated their time series of X, Y, and Z coordinates. The coordinates were then divided into two groups based on whether the upper and lower halves of the helixes (backbone atoms) were in contact ($\leq 0.5$ nm) with cholesterol (heavy atom) or not. Separately in each group, the distance from the average point of the group at each time frame (say $i^{th}$ frame) was calculated by:

$$d_i^2 = (x_i - x_g)^2 + (y_i - y_g)^2 + (z_i - z_g)^2,$$

where $x_i$, $y_i$, $z_i$ were the coordinates of the $i^{th}$ frame, and $x_g$, $y_g$, $z_g$ were the average values. The standard deviation of each group was then calculated by:

$$\sigma = \sqrt{\frac{1}{N_g} \sum_{i=1}^{N_g} d_i^2}$$

The average standard deviation of different simulations was calculated as a weighted average depending on the number of frames ($N_g$) of the group in each simulation.

## Two-dimensional (2D) number density map

The 2D number density maps were calculated using the g_densmap tool of GROMACS. The two bilayer leaflets were calculated separately. The output was then processed (using an in-house script) to normalize the maximum number density to one. We calculated the 2D number densities of cholesterol (non-hydrogen atoms) and β₂AR (backbone atoms of transmembrane region) separately.

## Cholesterol occupancy time per residue

A residue of β₂AR was considered to be in contact with cholesterol, when any of its non-hydrogen atoms was within ≤0.5 nm of any heavy atom of cholesterol. The total occupancy time was then normalized over the entire length of a simulation, i.e., an occupancy time of one means that the particular residue of β₂AR was in contact with cholesterol throughout the simulation, whereas a value of zero means no contact. The calculated total occupancy time per residue of β₂AR was mapped onto the receptor's surface to highlight the regions of β₂AR involved in cholesterol binding.

## Sequence alignment at cholesterol-binding sites

We analyzed amino acid sequences of $\beta_2AR$ orthologues from the available databases. We used the PhylomeDB server (http://phylomedb.org/) (*Huerta-Cepas et al., 2014*) for finding orthologues and Clustal Omega (http://www.ebi.ac.uk/Tools/msa/clustalo/) (*Sievers et al., 2011*) for sequence alignment. The amino acid residues of $\beta_2AR$ segments constituting the cholesterol binding sites as obtained from our simulations were used for the set of sequences obtained (*Figure 2—figure supplement 2*, *Figure 2—figure supplement 3*, *Figure 2—figure supplement 4*). The sequences in question belong to diverse species, such as insects, fish, birds, reptile, mammals, etc.

## Lipid tail order parameter

The order parameter of lipid acyl chains was calculated using :

$$S_{CD} = \left\langle \frac{3}{2} \left( \cos^2 \theta_i \right) - \frac{1}{2} \right\rangle$$

where $\theta_i$ is the angle between a C-D bond (C-H in simulations) of the $i^{th}$ carbon atom and the bilayer normal. The angular brackets denote averaging over time and molecules in a bilayer.

## Bilayer thickness

Bilayer thickness was defined as the distance between the average planes formed by phosphorous atoms in the two bilayer leaflets. We used the g_lomepro tool (*Gapsys et al., 2013*) to generate the 2D distribution of bilayer thickness.

## Lifetime of cholesterol binding

For the calculation of the lifetime of cholesterol bound to the cholesterol interaction sites on the receptor surface, we first monitored the binding/unbinding events of each individual cholesterol molecule along the simulation trajectory. A cholesterol molecule was considered bound when any of its heavy atoms came within $\leq 0.5$ nm from an interaction site. To define the three major interaction sites on the $\beta_2AR$ surface, we used the amino acid residues (with contact fraction $\geq 0.4$) as shown in *Figure 2—figure supplement 2*. The time series was then additionally smoothed (over one ns time windows) to discard very rapid 'leave and return' motions of cholesterol that take place due to thermal fluctuations. Given that lateral diffusion of lipids at the protein surface is very slow, and the lipids essentially do not move at all during a 1-ns time window, these fluctuations were then taken care of by the smoothing procedure. We then calculated the normalized time correlation function (to describe the time-dependent probability of cholesterol that is next to the receptor to stay in contact with the receptor) over all individual cholesterol binding/unbinding events occurred in all independent simulation trajectories for all cholesterol molecules present in a system at a given cholesterol concentration (*Arnarez et al., 2013*; *Horn et al., 2014*).

## Equilibration and error bar estimation associated with analysis

For all analysis to measure time-averaged properties, the first 100 ns of production simulations were excluded from the calculation. Error bars were estimated through standard error, calculated by dividing the standard deviation of a given data set with the square root of its sample size (*Manna et al., 2015*; *Kulig et al., 2014*). We used the g_analyze tool of GROMACS for error estimation.

## Acknowledgements

Dr. Maria Kalimeri is thanked for discussions. CSC – Finnish IT Center for Scientific Computing (Espoo, Finland) and PRACE through the HLRS High-Performance Computing Center (Stuttgart, Germany) are acknowledged for computer resources. European Research Council (Advanced Grant project CROWDED-PRO-LIPIDS) and the Academy of Finland (Centre of Excellence program) are thanked for financial support.

# Additional information

## Funding

| Funder | Grant reference number | Author |
|---|---|---|
| European Research Council | 290974 | Moutusi Manna<br>Waldemar Kulig<br>Tomasz Rog<br>Ilpo Vattulainen |
| Suomen Akatemia | 272130 | Moutusi Manna<br>Joona Tynkkynen<br>Matti Javanainen<br>Waldemar Kulig<br>Tomasz Rog<br>Ilpo Vattulainen |

The funders had no role in study design, data collection and interpretation, or the decision to submit the work for publication.

## Author contributions

MM, Conception and design, Acquisition of data, Analysis and interpretation of data, Drafting or revising the article; MN, JT, Acquisition of data, Analysis and interpretation of data; MJ, Acquisition of data, Drafting or revising the article, Contributed unpublished essential data or reagents; WK, Analysis and interpretation of data, Drafting or revising the article; DJM, Conception and design, Drafting or revising the article; TR, IV, Conception and design, Analysis and interpretation of data, Drafting or revising the article

## Author ORCIDs

Moutusi Manna, http://orcid.org/0000-0001-9472-1594
Matti Javanainen, http://orcid.org/0000-0003-4858-364X
Ilpo Vattulainen, http://orcid.org/0000-0001-7408-3214

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
