## [Decision Letter]

Thank you for submitting your article "Mechanism of allosteric regulation of β2-adrenergic receptor by cholesterol" for consideration by *eLife*. Your article has been favorably evaluated by Arup Chakraborty (Senior Editor) and two reviewers, one of whom, Nir Ben-Tal (Reviewer #1), is a member of our Board of Reviewing Editors.

The reviewers have discussed the reviews with one another and the Reviewing Editor has drafted this decision to help you prepare a revised submission.

Summary:

The effects of membrane lipids on the structural and dynamic properties of membrane-bound proteins, as well as on their biological function, has been the subject of numerous studies. Cholesterol constitutes a particularly interesting example, as it has a complex effect on membrane structure and is also known to bind specifically to many membrane proteins. One such case is the β2-adrenergic receptor (β2-AR), to which cholesterol molecules have been shown to bind specifically. Moreover, cholesterol has been found to improve the stability, ligand binding, and signaling properties of the β2-AR. However, the mechanisms underlying these effects have never been explained in detail, and it is unclear whether cholesterol acts directly or by changing membrane properties like thickness or order. Aside of the academic interest in understanding how cholesterol modulates GPCR action, this issue is also important for the pharmaceutical industry. GPCRs constitute a major target for pharmaceutical drugs and there is a growing interest in finding molecules that can modulate GPCR activity by binding to allosteric sites.

In the current project, Vattulainen et al. studied the effect of cholesterol on the structural and dynamic properties of the β2-AR using extensive MD simulations. The simulations predicted three main cholesterol binding sites. The first (IC1) is in agreement with the crystal structures of the β2-AR, and is at the general area of a known conserved cholesterol-binding motif (CCM). The second binding site (IC2) is undocumented, and the third (EC1) is in agreement with the crystal structure of the adenosine 2A receptor. While the validity of EC1 and IC2 as specific cholesterol binding sites is yet to be confirmed, the fact that the well-documented IC1 has also been predicted by the same simulations is encouraging. Furthermore, EC1 and IC2, if valid, can be used as potential target sites for GPCR-specific drugs.

The simulations also predicted two cholesterol-induced effects on the β2-AR. The first is a general restriction of the inherent dynamics of the protein. This effect was not observed when the general properties of the membrane were changed in the absence of cholesterol, and thus, the authors concluded that this effect is specific. The second cholesterol-induced effect predicted by the simulation appeared in the second binding site, IC2. There, cholesterol was predicted to push the intracellular end of TM6 more towards the core of the helical bundle, and prevented the outward movement of this helix. This effect is particularly interesting, as the outward movement of TM6 is associated with GPCR activation, and creates the binding site for the receptor's cognate G-protein. The restriction of TM6 movement by cholesterol is a potentially important allosteric effect, which again, can be used to modulate GPCR activity.

This is an important project and the manuscript reads well, however, a number of outstanding issues, listed below, should be addressed before decision can be made about publication.

Essential revisions:

1) The general restriction of β2-AR dynamics by cholesterol is rather obvious considering the rigid structure of cholesterol; this rigidity would restrict the dynamics of any molecule bound to cholesterol, be it a neighboring membrane lipid or a protein. Having said that, the fact that cholesterol analogues had a weaker effect on protein dynamics despite their rigid structure suggests that additional factors are in play. Perhaps this issue could be explored in greater detail to decipher the energy determinants and physicochemical underlines.

2) A discussion of the effect of cholesterol binding to the first binding site (IC1) is missing. This site resides in a cleft created by TMs 1-4 and contains the conserved cholesterol consensus motif (CCM). The conservation of the CCM has implicated it as a possible allosteric site in class A GPCRs. Since the current study focusses on possible allosteric effects of cholesterol, the neglect of IC1 in the analysis of the results seems odd. Perhaps the authors can extract this information from the existing simulations.

3) Furthermore, the results could be correlated with evolutionary data (e.g., using ConSurf). The anticipation is that biologically relevant binding sites would be shared among other GPCRs (orthologues at the very least), which implies that the binding residues should be evolutionarily conserved.

4) Figure 1—figure supplement 1: That the shape of the distances distribution changes with cholesterol concentration in a non-monotonic manner is of concern. Maybe in spite of the long simulation time the results are still not converged in all ranges?

5) This has implications on the main research question here, i.e., whether cholesterol affects the conformational changes of the receptor directly or via general effect on membrane properties. The authors argue that the distribution of receptor' conformations when cholesterol binds the receptor directly (Figure 1) is markedly different in comparison to when it does not (Figure 3). However, to me the difference is small, and in view of the non-monotonic behavior mentioned above, the conclusion might be erroneous.

6) While the focus here is on inactivation it would be nice to show also activation for completeness.

7) In the Introduction – we would value a little more background on what is known of the effect of cholesterol on GPCR and specifically β2AdR function. The Introduction says that cholesterol likely interacts with GPCRs and 'has been shown to influence the ligand binding and signaling properties of β2AR'. This is a bit vague given this underlies the whole of the study presented here. Or perhaps not much is known experimentally, in which case to what extent can one formulate a clear hypothesis to be tested via simulation?

8) Introduction, last paragraph. The 'physiological' concentration of cholesterol is given as 10 mol%; Sampaio et al. says cholesterol concentration in e.g. epithelial cell membranes is more like 25 to 30 mol% (Sampaio et al., 2011, PNAS).

9) Subsection “Membrane-mediated interactions not the key”: the authors show quite conclusively that the effects of cholesterol on the conformational dynamics of the receptor and are not due simply to a change in the physical state of the surrounding bilayer. A clinching test would be to place cholesterol at the binding sites (perhaps by taking a snapshot from the high cholesterol simulation), then place the receptor/cholesterol complex in a cholesterol free membrane and see how the conformational dynamics of the protein change as the cholesterol is release. Has this been done? From the discussion of binding lifetimes (subsection “Binding lifetime depends on cholesterol”), the bound cholesterols might dissociate on a 0.1 µs timescale. Indeed, is this the simulation in Figure 1—figure supplement 1 (it is not clear – Table 1 is a bit impenetrable)? And if so, what is the time course for unbinding of the cholesterols? This needs to be explored/explained in more detail.

10) Subsection “Cholesterol analogues interact with β2AR”. Are there any experimental data for the specificity of the cholesterol effects on β2AdR function?

11) The Discussion could perhaps be a bit tighter – to some extent it re-iterates what has been said earlier.

---

## [Author Response]

*Essential revisions:*

*1) The general restriction of β2-AR dynamics by cholesterol is rather obvious considering the rigid structure of cholesterol; this rigidity would restrict the dynamics of any molecule bound to cholesterol, be it a neighboring membrane lipid or a protein. Having said that, the fact that cholesterol analogues had a weaker effect on protein dynamics despite their rigid structure suggests that additional factors are in play. Perhaps this issue could be explored in greater detail to decipher the energy determinants and physicochemical underlines.*

Our results suggest that the restriction to β_2_AR dynamics arises from the binding of cholesterol to specific interaction sites on the receptor surface. Cholesterol analogues, although being structurally very similar to cholesterol, induce weaker effects. We quantified this as follows.

We found that while both cholesterol and its analogues have a rigid ring structure, their different chemical compositions (such as the esterified succinic acid group in cholesteryl hemisuccinate (CHS) compared to the hydroxyl group in cholesterol (at the 3β position), or the additional hydroxyl groups in oxysterols not found in cholesterol) influence their occupancy as well as the strength of binding to the specific interaction sites. To explore this in more detail, we calculated the total van der Waals (vdW) interaction energy as well as the number of contacts between cholesterol and β_2_AR, when cholesterol is in the IC1, IC2, or EC1 binding site (please see Table 2). Similar calculations were carried out for the cholesterol analogues. Given that the van der Waals energy is a short-range interaction, it reflects how strongly the two molecules are in contact and therefore serves as an excellent measure for the gravity of lipid-protein binding in the binding site. The results presented in Table 2 show without doubt that the van der Waals interaction of cholesterol with β_2_AR is substantially stronger than in the case of CHS or the oxysterols.

The new results show that among the three major interaction sites, the binding of CHS at IC1 is much weaker than that of cholesterol. At IC2 the strength of interaction of CHS and cholesterol are comparable. Meanwhile, the extracellular EC1 site remains unoccupied by CHS indicating the binding energy to be low. As to the two oxysterols considered in our study, 4β-OH-Chol interacts with β_2_AR only at EC1 and the interaction is weak. 27-OH-Chol binds at EC1 as tightly as cholesterol, but its interaction at the two other binding sites (IC1 and IC2) is much weaker than in the case of cholesterol. Concluding, CHS interacts at IC2 as strongly as cholesterol but its interactions at IC1 and EC1 are negligible compared to those of cholesterol. The oxysterol 27-OH-Chol interacts at EC1 as strongly as cholesterol but its interactions at IC1 and IC2 are negligible compared to those of cholesterol. The oxysterol 4β-OH-Chol does not interact with β_2_AR to a significant degree.

Consistently, we earlier found that CHS restricts the dynamics of the β_2_AR G protein binding site similarly as cholesterol (L_G_inFigure 2—figure supplement 9), which supports our hypothesis that the binding of cholesterol or cholesterol-like molecules at IC2 (composed of intracellular ends of H5 and H6) suppresses the dynamics at the G protein binding site by suppressing the dynamics of respective helices. Similarly as EC1 remains completely unoccupied by CHS, it cannot restrict the dynamics of the extracellular ligand-binding site of the receptor (L_L_in Figure 2—figure supplement 9).

These data can be considered in the context of molecular structures. In CHS, the difference compared to cholesterol is the additional chain coupled to cholesterol via an ester bond (Figure 2—figure supplement 6). Based on our results this additional chain does not interfere binding at IC2, but it does interfere the binding at IC1 and EC1. In 27-OH-Chol, the oxidation has taken place in the short acyl chain that is the terminal subunit of the molecule. This does not interfere the binding at EC1 but does alter the binding at IC1 and IC2. Finally, in 4β-OH-Chol, the oxidation has taken place in the rigid steroid moiety, making the α-side of the molecule rougher. In cholesterol, the α-side is exceptionally flat. Given this change in surface roughness, and the importance of the surface-surface contact in lipid-β_2_AR binding in the binding site, it is quite obvious why this oxysterol does not bind with any of the cholesterol binding sites (IC1, IC2, EC1).

The new results confirm that the binding of cholesterol or cholesterol-like molecules at the specific interaction sites restrict the conformational fluctuation of β_2_AR – tighter binding suppresses β_2_AR dynamics considerably, and the effect of cholesterol is the strongest. The new results also provide a more solid structural basis to understand why the binding of cholesterol with β_2_AR is superior compared to CHS or the oxysterols.

Based on the above, we have done the following revisions and additions:

We have discussed the above ideas (Section: Cholesterol analogues interact with β_2_AR);

We have presented Table 2 and discussed its results.

*2) A discussion of the effect of cholesterol binding to the first binding site (IC1) is missing. This site resides in a cleft created by TMs 1-4 and contains the conserved cholesterol consensus motif (CCM). The conservation of the CCM has implicated it as a possible allosteric site in class A GPCRs. Since the current study focusses on possible allosteric effects of cholesterol, the neglect of IC1 in the analysis of the results seems odd. Perhaps the authors can extract this information from the existing simulations.*

The X-ray crystal structure (PDB id: 3D4S) of β_2_AR has established a specific cholesterol-binding site between the intracellular segments of helices (H) 1, 2, 3, and 4 (Gilchrist, 2010). The given study defines a cholesterol consensus motif (CCM) [4.39-4.43(R,K)]—[4.50(W,Y)]—[4.46(I,V,L)]—[2.41(F,Y)], which predicts cholesterol binding for 44% of human class A receptors (Gilchrist, 2010). Our present simulation study reproduced cholesterol binding at this interaction site (named in our paper as IC1) and satisfied the condition of strict-CCM binding, where the conserved residues (R151^4.43^, W158^4.50,^I154^4.46^, andY70^2.41^) participated in cholesterol binding (Figure 6). The close agreement between our simulation results and the experimental prediction strengthens the view that the simulation approach we have used is valid.

Author response image 1.Specific cholesterol binding site in β_2_AR with CCM displayed with side chain positions of conserved amino acid residues, as found in (**A**) the crystal structure (1) and (**B**) during our simulation.In the simulation snapshot, residues are colored according to their strength of interaction with cholesterol (red represents the weakest and blue represents the strongest interaction).**DOI:**
http://dx.doi.org/10.7554/eLife.18432.026

The physiological relevance of specific cholesterol binding at IC1 is not yet fully understood. In the article describing the crystal structure, the authors used the occluded surface area method to analyze internal packing of helices (Gilchrist, 2010). The study showed that cholesterol increases the packing of H4, indicating decreasing mobility (Gilchrist, 2010). The effect was suggested to be due to the binding of cholesterol at IC1, where H4 establishes major binding interactions (Figure 6). From the simulation trajectory we calculated the standard deviation of H4 fluctuations, when IC1 is either occupied or unoccupied by cholesterol (Figure 7). Our results show that the fluctuations of H4 around its average position are more suppressed when cholesterol is bound to IC1. This result is in agreement with experimental findings. While H4 is not one of the strongest points of β_2_AR fold, its decreased mobility may account for the enhanced stability of the receptor. We did not observe similar stabilizing effects of cholesterol binding at IC1 on the helices H1-H3. Finally, a recent simulation study reported a correlation between cholesterol occupancy at IC1 and β_2_AR dimerization (Lefkowitz, 2000). This study showed that cholesterol occupancy at H4 restricts its participation in dimer interface formation and subsequently stabilizes the dimer interface with H1 and H2 instead of H4 and H5 (Lefkowitz, 2000).

Author response image 2.For the time-dependent distance betweenH4 and its average position, as the H4 helix fluctuates around its average location, shown here are results for the standard deviation of the distance fluctuations.Data are given for cases, where IC1 is occupied (blue) or unoccupied (orange) by cholesterol.**DOI:**
http://dx.doi.org/10.7554/eLife.18432.027

We move on to discuss if there is any impact of IC1 on the observed conformational change of β_2_AR reported in our work. To decouple the effect of IC1 from the two other binding sites, we consider the following cases. Our results show that CHS interacts at IC2 as strongly as cholesterol (Table 2). In systems with 10 mol% CHS, we observed a high CHS density at IC2, but not at IC1 or EC1 (Figure 2—figure supplement 8). In the system with 40 mol% CHS, in addition to a high CHS density at IC2, IC1 is also occasionally occupied by CHS, while EC1 remains unoccupied (Figure 2—figure supplement 8). In either of these cases, CHS restricts the dynamics of the G protein-binding site similarly to the extent of cholesterol (Figure 1—figure supplement 1 and Figure 2—figure supplement 9), irrespective of whether IC1 is occupied or unoccupied by cholesterol. The result indicates that sterol occupancy at IC2, but not at IC1, is crucial for controlling the dynamics of the G protein binding site, calculated by the displacement of the intracellular part of H6. This conclusion is further supported by the observation that in the presence of 5 mol% cholesterol, β_2_AR exhibits large fluctuations at the G protein binding site (Figure 1—figure supplement 1 and Figure 8), where IC1 is occupied by cholesterol but IC2 remains unoccupied (Figure 2—figure supplement 5(top panel)). In bilayers with CHS, no CHS was observed at the extracellular binding site EC1. Consequently, in these systems β_2_AR shows large fluctuations at the extracellular ligand binding site, irrespective of the high CHS occupancy at IC2 and also at IC1 for systems with 40 mol% CHS.

Altogether these results suggest that IC1, located at the intracellular surface cleft between H1-H4, does not show much influence on the mobility of H5 and H6, which play a major role in determining the conformational dynamics of β_2_AR.

Based on the above, we have done the following revisions and additions:

We have discussed the above ideas (Section: Discussion);

We have added Figure 2—figure supplement 10and discussed this new data;

We have included two new videos (Video 1 and Video 2 (related to Figure 5)).

*3) Furthermore, the results could be correlated with evolutionary data (e.g., using ConSurf). The anticipation is that biologically relevant binding sites would be shared among other GPCRs (orthologues at the very least), which implies that the binding residues should be evolutionarily conserved.*

Our simulations suggest three main binding sites of cholesterol on β_2_AR (Figure 2). One of these sites (IC1), located at the intracellular surface cleft between H1-4, matches very well the recently reported crystal structure of β_2_AR (Gilchrist, 2010). This site contains a cholesterol consensus motif (CCM) with the sequence [4.39-4.43(R,K)]—[4.50(W,Y)]—[4.46(I,V,L)]—[2.41(F,Y)]. As also discussed in the previous point, our simulations reproduce interactions between these conserved amino acid residues and cholesterol (Figure 6) that are important in binding cholesterol at IC1.44% of human class A receptors are predicted to bind cholesterol at the same site as β_2_AR (see Figure 5 and Table 2 of Gilchrist, 2010). In addition to IC1, our simulations proposed two other cholesterol interaction sites (IC2 and EC1) on β_2_AR. EC1 is in agreement with the location of cholesterol found in the crystal structure of the adenosine receptor A_2A_AR (Manglik and Kobilka, 2014).

As suggested by the reviewers, in order to examine whether our predicted β_2_AR cholesterol binding sites are conserved during the evolution of the receptor, we analyzed amino acid sequences of β_2_AR orthologues from available databases (Figure 2—figure supplement 2, Figure 2—figure supplement 3, Figure 2—figure supplement 4). We used the PhylomeDB server (http://phylomedb.org/) (Kobilka, 2013) for finding orthologues and Clustal Omega (http://www.ebi.ac.uk/Tools/msa/clustalo/) (Cherezov et al., 2007) for sequence alignment. The amino acid residues of β_2_AR segments constituting the cholesterol binding sites as obtained from our simulations are used for the set of sequences obtained (Figure 2—figure supplement 2, Figure 2—figure supplement 3, Figure 2—figure supplement 4). The following sequences belong to diverse species, such as insects, fish, birds, reptile, mammals, etc. The alignments show that the cholesterol binding residues of the three interaction sites are conserved in most of the species. Therefore, it appears that these cholesterol-binding sites are evolutionarily conserved in β_2_AR.

Based on the above, we have done the following revisions and additions to the paper:

We have discussed the above ideas in the main paper (Section: Specific binding of cholesterol);

We have added Figure 2—figure supplement 2, Figure 2—figure supplement 3, Figure 2—figure supplement 4, where we provide this new data in full.

*4) Figure 1—figure supplement 1: That the shape of the distances distribution changes with cholesterol concentration in a non-monotonic manner is of concern. Maybe in spite of the long simulation time the results are still not converged in all ranges?*

We are afraid that there was a misunderstanding of the data shown in the original manuscript. We consider this possibility in the end of this point, however let us first discuss the data and show that their cholesterol concentration dependence is systematic/monotonic, as expected.

In Figure 8 we depict the distributions of L_L_ (width of the ligand binding site) and L_G_ (width of the G protein interface) distances separately for each individual trajectory and for all the different cholesterol concentrations (0-40 mol%). These data correspond to the L_L_ vs. L_G_ distance distributions shown in Figure 1—figure supplement 1. It is apparent from Figure 8 that the distributions of both L_L_ and L_G_ are broad in the absence of cholesterol, and also at a very low cholesterol concentration (2 mol%), indicating that for low cholesterol concentrations the β_2_AR receptor undergoes continuous conformational fluctuations between “wide ligand-binding site/narrow G protein-binding site” and “narrow ligand-binding site/wide G protein binding site”. However, the distributions become more and more narrow with increasing cholesterol concentration. At cholesterol concentrations above about 10 mol%, one finds sharper distributions with a single peak, reflecting the confinement of the receptor in a particular conformational state. Moreover, Figure 8 shows that in individual simulations for a given cholesterol concentration, the receptor samples essentially similar states, indicating quite complete convergence of the present simulation results. The results therefore confirm a systematic/monotonic effect of cholesterol concentration on the β_2_AR conformational distribution.

Further, we have binned the 2D conformational distance data (shown, e.g., in Figure 1in the main text) to a 2D histogram, and then counted its ‘area' by calculating the number of bins that had a non-zero value. The results depicted in Figure 9 show that for increasing cholesterol concentration, the area covered by the (L_L_, L_G_) conformational distribution decreases, in line with the data shown in Autghor response Figure 3. To make sure that this result is not compromised by bins where the value of the probability of being in a given (L_L_, L_G_)-state is very low, we repeated the analysis by disregarding the bins whose occupancy was less than 1% of the maximum value in the 2D distribution. The latter calculation (cutoff 0.01 x P_max_ shown in Figure 9) does not affect the result: the effect of cholesterol is systematic (Figure 9).

Therefore, the data presented here show that the conformational distribution of the receptor determined as a function of L_L_ and L_G_ distances changes in a systematic/monotonic manner with increasing cholesterol concentration from 0 to 40 mol%. These concentration values correspond to the distance distributions presented in panels A-E(earlier B-F) of Figure 1—figure supplement 1.

Panel F(earlier A) of Figure 1—figure supplement 1is an exception, however here the initial configuration of the simulated system was distinctly different compared to the other simulated systems. Figure 1—figure supplement 1(earlier 1A) describes a situation, where 8 cholesterol molecules were initially (at time zero of the simulation) bound at the 8 interaction sites on β_2_AR that were predicted by our simulations and previous experimental data, while the rest of the membrane system had no cholesterol. Therefore, in this system, at large times in the multi-microsecond scales, all or at least most of the cholesterol molecules were no longer bound to β_2_AR and instead they resides in the membrane far from the receptor – therefore at long times this system describes cholesterol-poor conditions. We consider that since this background was not described to a sufficient degree in the original paper, the reviewers unfortunately misunderstood the situation described by this system. A detailed discussion of this system and its results are given in the point 9 below.

Based on the above, we have clarified the concerns and questions presented. However, we concluded that the new results (Figure 8 and Figure 9) do not add significant new insight to the paper; therefore, they have not been included to the paper.

Author response image 3.Distributions of L_L_ and L_G_ distances from individual trajectories (shown in different colors) for various cholesterol concentrations.**DOI:**
http://dx.doi.org/10.7554/eLife.18432.028

Author response image 4.Area in the 2D histogram visited by the receptor conformations.The bin edge length was set to 0.1 Å in both dimensions.**DOI:**
http://dx.doi.org/10.7554/eLife.18432.029

*5) This has implications on the main research question here, i.e., whether cholesterol affects the conformational changes of the receptor directly or via general effect on membrane properties. The authors argue that the distribution of receptor' conformations when cholesterol binds the receptor directly (Figure 1) is markedly different in comparison to when it does not (Figure 3). However, to me the difference is small, and in view of the non-monotonic behavior mentioned above, the conclusion might be erroneous.*

Please see our answer above regarding the “non-monotonic” behavior. Figure 1 represents the distribution of the receptor’s conformation in the presence of 10 mol% cholesterol when cholesterol binds to the receptor directly. As we discussed in the earlier point, here the distribution is restricted to one particular conformation with L_L_ ~ 13 Å and L_G_ ~ 9.5 Å. The receptor samples a similar conformational space at higher cholesterol concentrations (Figure 1—figure supplement 1 and Figure 8).

On the other hand, in the absence of cholesterol, β_2_AR samples a much broader conformational space (Figure 1 and Figure 8) with L_L_ ranging between ~11-17 Å and L_G_ ranging between ~7-13 Å).

Importantly, the broad conformational distribution of β_2_AR found under cholesterol-poor conditions is also observed in cholesterol-free bilayers whose physical properties are consistent with cholesterol-rich bilayers (but without cholesterol). First, cholesterol is known to increase membrane thickness significantly. We induced this effect with long-chain PC lipids that give rise to a thick membrane and found (see Figure 4) (earlier Figure 3) that in this system L_L_ fluctuates strongly between ~11-16.5 Å and L_G_ varies between ~7-12.5 Å. This implies that a thick membrane does not induce the restricted conformational behavior we found in cholesterol-rich bilayer (Figure 1). Second, cholesterol is known to order lipid membranes significantly. To test this membrane-mediated effect, we used pyrene instead of cholesterol to increase membrane order to a level that matches the membrane order found with 10 mol% of cholesterol. The results (Figure 4) show L_L_ to range between ~9-17.5 Å and L_G_ to vary between ~7-13.5 Å, therefore the receptor’s conformational fluctuations match the data found in the absence of cholesterol. This implies that membrane-mediated interactions do not cause the restricted conformational behavior we found with cholesterol.

Altogether, the data show that when cholesterol concentration is increased from low to high (from about zero to 10 mol%), the conformational behavior of β_2_AR is altered significantly. At small cholesterol concentrations, the ligand and the G protein binding sites of β_2_AR fluctuate between wide and narrow states, while at high cholesterol concentrations the β_2_AR conformation (and the ligand and the G protein binding sites) are fixed to a single state. These trends are systematic and consistent as we increase cholesterol concentration from zero to 40 mol%. These data were consistent when we repeated the simulations for every cholesterol concentration many times.

Based on the above, we have done the following revisions:

We have clarified the concern as to the assumed non-monotonic behavior (Figure 1—figure supplement 1, Table 1);

We have included two new videos (Video 1 and Video 2 (related to Figure 5)).

*6) While the focus here is on inactivation it would be nice to show also activation for completeness.*

Studying activation, i.e., transition from the inactive to the active sate β_2_AR is beyond the present work given that the timescale of the activation process is likely milliseconds. Nonetheless, we followed the reviewers’ recommendation and carried out an additional extensive set of simulations for the active-state receptor.

We studied the effect of cholesterol on the conformation of the active state β_2_AR in apo form and in the absence of the G protein (Hanson et al., 2008). We conducted replicate unbiased atomistic MD simulations of the active state β_2_AR embedded in a DOPC bilayer with 0 and 40 mol% cholesterol (three independent 2.5 μs simulations for each system, see Table 1). In the active form, the intracellular end of H6 is splayed outward from the helical bundle providing room for the G protein (Figure 10).

In the absence of cholesterol (0 mol%), we observed inward swinging of H6 towards H3 (observed in 2 out of 3 replicas). As shown in Figure 10, the intracellular end of H6 spontaneously approaches H3 with L_G_ dropping from 18.97 Å in the starting active conformation to ~11.5 Å that matches the crystallographically observed inactive conformation of β_2_AR (L_G_ ~11 Å). Such spontaneous deactivation of the receptor in the absence of the intracellular binding partner is in agreement with recent simulations (Rasmussen et al., 2011; Nygaard et al., 2013) and experimental studies (Dror et al., 2009).

In the presence of cholesterol (40 mol%), we observed that the active-like open conformation (without G protein) is stable during the simulations (Figure 3—figure supplement 1). As shown in Figure 10,the L_G_ value remains stable ~16.5 Å and no deactivation is observed unlike in cholesterol-free systems. Interestingly, we found a high cholesterol density at the IC2 interaction site located at the cleft between the intracellular segments of H5-H6 (Figure 10). Cholesterol bound at IC2 acts as a spacer between H5-H6 and restricts the movement of H6, thereby stabilizing the open active-like conformation of the receptor (Figure 10).

These results for the active state β_2_AR support our previous conclusion that cholesterol restricts the conformational dynamics of the receptor by binding at specific interaction sites. Altogether, our results (based on both inactive and active-state β_2_AR conformations) suggest that specific interactions of cholesterol with β_2_AR govern changes between the receptor’s different conformational states, which in turn affect receptor activation. Moreover, the results highlight the importance of IC2: cholesterol binding at IC2 in both inactive and active conformational states of β_2_AR as found in our simulations indicates the significant biological relevance of this interaction site.

Based on the above, we have done the following revisions and additions:

We have discussed the above ideas in the paper (Sections: Cholesterol restricts β_2_AR conformation; Discussion) and shown as a new Figure 3;

We have added Figure 3—figure supplement 1.

Author response image 5.Cytosolic view of β_2_AR (**A**) in the beginning of simulation (active state) as well as in representative simulation snapshots in (**B**) a DOPC bilayer and (**C**) in the presence of 40 mol% cholesterol.The dotted line represents the distance between the Cα atoms of R131^3.50^–E268^6.30^ (defined as L_G_) used to measure the fluctuation at the G protein binding site. (**D**) Simulation snapshot (in the presence of 40 mol% cholesterol) showing cholesterol binding at the interaction sites of β_2_AR. (**E**) The time evolution of L_G_ in systems with 0 (light red) and 40 mol% cholesterol (light blue). Corresponding 50-point running averages are shown in dark colors. (**F**) 2D number densities of cholesterol (Chol) around β_2_AR (cytosolic view). The transmembrane regions of β_2_AR are shown in gray scale (the darker the color, the higher is the number density), and they are numbered accordingly to show the locations of the individual helices (H1-H7).**DOI:**
http://dx.doi.org/10.7554/eLife.18432.030

*7) In the Introduction we would value a little more background on what is known of the effect of cholesterol on GPCR and specifically β2AdR function. The Introduction says that cholesterol likely interacts with GPCRs and 'has been shown to influence the ligand binding and signaling properties of β2AR'. This is a bit vague given this underlies the whole of the study presented here. Or perhaps not much is known experimentally, in which case to what extent can one formulate a clear hypothesis to be tested via simulation?*

β_2_AR belongs to the family of class A GPCRs. GPCRs belonging to this class show a high structural similarity and functional diversity. The literature reporting on the specific functional role of cholesterol and other lipids is extensive (Ozcan et al., 2013; Dror et al., 2011). Reportedly cholesterol has been shown to affect the conformation (Oates and Watta, 2011; Lingwood and Simons, 2010) and function (Ozcan et al., 2013; Oates and Watta, 2011;Allen, Halverson-Tamboli and Resenick, 2007; Contreras et al., 2012; Coskun et al., 2011) of many GPCRs. Experimental data have been published that cholesterol binds specifically to β_2_AR (Gilchrist, 2010; Neale et al., 2015), and it has been experimentally shown that cholesterol binding to β_2_AR changes its structural properties (Gilchrist, 2010; Dawaliby et al., 2016). Since structure and function of GPCRs are closely related one may thus expect that cholesterol binding specifically to β_2_AR changes also the functional properties of the receptor. Indeed a few experimental papers have been published which indicate that cholesterol also has a functional role (Paila and Chattopadhyay, 2009; Gimpl, Burger and Fahrenholz, 1997). Thus, in our study we wanted to understand a molecular dynamics basis where and how cholesterol binds to β_2_AR and which are the possible structural and functional effects.

Based on the above, we have done the following revisions:

We have revised the Introduction accordingly, together with related new references.

*8) Introduction, last paragraph. The 'physiological' concentration of cholesterol is given as 10 mol%; Sampaio et al. says cholesterol concentration in e.g. epithelial cell membranes is more like 25 to 30 mol% (Sampaio et al., 2011, PNAS).*

We have removed the term “physiological” and modified the manuscript accordingly.

*9) Subsection “Membrane-mediated interactions not the key”: the authors show quite conclusively that the effects of cholesterol on the conformational dynamics of the receptor and are not due simply to a change in the physical state of the surrounding bilayer. A clinching test would be to place cholesterol at the binding sites (perhaps by taking a snapshot from the high cholesterol simulation), then place the receptor/cholesterol complex in a cholesterol free membrane and see how the conformational dynamics of the protein change as the cholesterol is release. Has this been done? From the discussion of binding lifetimes (subsection “Binding lifetime depends on cholesterol”), the bound cholesterols might dissociate on a 0.1 µs timescale. Indeed, is this the simulation in Figure 1—figure supplement 1 (it is not clear – Table 1 is a bit impenetrable)? And if so, what is the time course for unbinding of the cholesterols? This needs to be explored/explained in more detail.*

The reviewers correctly sensed that we have already done this test. Let us here explain this matter more clearly.

We performed simulations (3 replicas) where 8 cholesterol molecules were initially placed at the binding sites of β_2_AR that were predicted by our simulations (snapshot in Figure 5—figure supplement 1); then we placed the receptor/cholesterol complex in a cholesterol-free membrane (“Chol-Bound” trajectory in Table 1). The conformational distribution of β_2_AR is shown in Figure 1—figure supplement 1(earlier 1A, as correctly pointed out by the reviewer in a previous point). Figure 5—figure supplement 1 shows the time profile of unbinding events of cholesterol. From a majority of the binding sites cholesterol leaves very quickly in a timescale of tens to hundreds of nanoseconds (Figure 5—figure supplement 1), similarly to the short binding lifetime observed for cholesterol-poor systems (2 mol%, Figure 5). However, there are also sites where cholesterol stays for the entire simulation time (IC1 in repeats 2 and 3, and IC2 in repeats 1 and 3) or where cholesterol is released from the receptor in μs timescale (IC3 and EC3 in repeat 1).

Although for these systems the average cholesterol concentration is low (1.9 mol%), due to the setup of the initial system configuration, the concentration of cholesterol in the annular region is high when the simulation is started. Then, as some cholesterol molecules are released from the receptor during the course of the simulations, the concentration of cholesterols bound to the receptor decreases gradually but systematically. Given this, the early-time behavior of these systems corresponds to cholesterol-rich conditions while at long times the behavior corresponds to cholesterol-poor conditions.

When we analyzed the “Chol-bound” simulations (Figure 1—figure supplement 1), we found that the conformational behavior of β_2_AR changes from cholesterol-rich behavior (as in Figure 1—figure supplement 1) to cholesterol-poor behavior (as in Figure 1—figure supplement 1). However, presenting the time-dependence of the data in Figure 1—figure supplement 1is not particularly useful given that the sampling within each time window would be so limited that it would be hard to see the trends easily.

Based on the above, we have done the following revisions and additions to the paper:

We have discussed this data more concretely in the paper (Section: Cholesterol restricts β_2_AR conformation), where we also clarified Figure 1;

Table 1 has been clarified.

*10) Subsection “Cholesterol analogues interact with β2AR”. Are there any experimental data for the specificity of the cholesterol effects on β2AdR function?*

There are experimental data showing the specific role of cholesterol on β_2_AR on a functional (Paila and Chattopadhyay, 2009; Gimpl, Burger and Fahrenholz, 1997) and structural basis (Gilchrist, 2010; Dawaliby et al., 2016). The data presented in ref 19 on the influence of cholesterol manipulation on the signaling properties of β_2_AR show an inhibition of β_2_AR-associated signaling by increasing membrane cholesterol content. These are now briefly discussed in the article.

*11) The Discussion could perhaps be a bit tighter – to some extent it re-iterates what has been said earlier.*

We have followed the advice and revised/cut the Discussion to be more solid. The total length has not decreased though, since due to the additional simulations for the active-state receptor, the further discussion on the role of cholesterol interactions with the active-state β_2_AR had to be included in the Discussion, too.